# ME-Switch: A Memory-Efficient Expert Switching Framework for Large Language Models

## Abstract

The typical process for LLM's development involves pre-training a general foundation model on massive data, followed by fine-tuning on task-specific data to obtain a series of specialized experts. Serving these experts can pose significant memory challenges, as loading all experts onto devices is impractical, and frequent switching between experts in response to user requests can incur substantial I/O costs. Previous approaches decompose the expert weights as the pre-trained weights plus delta weights, followed by quantizing the delta weights using output channel-wise step sizes to reduce the model size. However, these methods overlook the fact that certain input channels of delta weights can cause significant quantization errors at extremely low bitwidths. To this end, we introduce ME-Switch, a memory-efficient expert switching framework tailored for serving multiple LLMs. To condense the number of bits required for describing the delta weights, we propose a salient-aware delta compression method that first identifies which input channels of delta weights are salient based on reconstruction error and then employs mixed-precision quantization that selectively quantizes non-salient input channels of delta weights to extremely low bits while keeping the salient ones intact, significantly reducing storage demand while maintaining performance. Extensive experiments show the promising memory efficiency and accuracy of ME-Switch. For example, when serving three models from the Mistral-7B family, ME-Switch reduces the model size by $2.04\times$ and maintains nearly lossless performance on instruction, mathematical reasoning, and code generation tasks. Furthermore, our method can efficiently serve 16 Mistral-7B models on an NVIDIA A100 GPU.

## 1 Introduction

Large language models (LLMs) such as GPT-4 (Achiam et al., 2023) and Gemini (Team et al., 2023), have achieved significant advancements in natural language processing (NLP). Following a pretrain-finetune paradigm (Touvron et al., 2023a;b; Dubey et al., 2024), these models are first trained on large-scale text datasets to develop a broad foundation of language understanding, enabling them to excel in tasks requiring common-sense knowledge. To acquire task-specific knowledge, these pre-trained models are further fine-tuned on specialized tasks, enabling their adaptation or alignment for diverse applications such as interactive agents (Touvron et al., 2023b; Jiang et al., 2023), code generation (Luo et al., 2023b; Lozhkov et al., 2024), and mathematical problem solving (Luo et al., 2023a; Lozhkov et al., 2024), demonstrating the remarkable versatility of LLMs. For instance, even high-capacity LLMs like the MoE model Mixtral-8x22B [1] are fine-tuned on instruction-following data to create specialized variants such as Mixtral-8x22B-Instruct-v0.1 [2], enhancing their ability to follow human instructions. While LLMs are powerful, fine-tuning for a specific task to enhance performance is generally more practical and efficient than multitask fine-tuning that often encounters conflicting objectives, mode collapse, and demands meticulous data mixing along with substantial training resources (Team et al., 2023; Touvron et al., 2023b). For example, DeepSeek-Coder-V2-Base (Zhu et al., 2024), a 236B-parameter MoE code model, is fine-tuned from DeepSeek-V2 (Liu et al., 2024a) to achieve significantly improved performance in the code domain but demonstrating reduced effectiveness in general question-answering tasks. This highlights the necessity of obtaining multiple task-specific LLMs.

---

[1] https://huggingface.co/mistralai/Mixtral-8x22B-v0.1
[2] https://huggingface.co/mistralai/Mixtral-8x22B-Instruct-v0.1

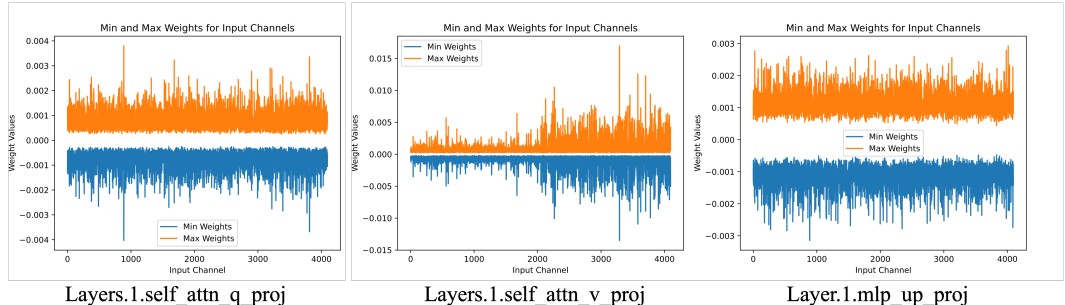

Figure 1: An illustration of the input channel-wise maximum and minimum values for the delta weights of Speechless-Code-Mistral-7B. The variability across input channels highlights that certain salient channels, irrespective of their magnitude, can cause significant quantization errors when quantized with ultra low-bitwidth, which underscores their critical role in preserving performance.

However, serving multiple models poses several major challenges. First, even with a relatively small number of models, the storage demands are significant due to the extensive number of parameters each model contains. For example, three LLaMA-2-70B models would collectively require over 384GB of storage, calculated as 128GB per model times three. Second, the substantial memory requirements of these models may make it impractical to load all of them into GPU memory simultaneously. While dynamically swapping model weights in and out of GPU memory as needed is feasible, the large size of the models makes this process slow and inefficient, significantly delaying response times and adversely affecting user experience.

To address the above challenges, existing methods (Liu et al., 2024b; Yao & Klimovic, 2023) decompose the weights of fine-tuned models into pre-trained weights and delta weights introduced during fine-tuning. While low-rank approximation (Hu et al., 2022) can compress these delta weights, it falls short for full fine-tuned models whose delta weights lack low-rank properties (Liu et al., 2024b; Lialin et al., 2024; Hao et al., 2024). As a result, prior work has adopted per-tensor quantization (Liu et al., 2024b) as an alternative, significantly reducing storage needs and facilitating efficient sharing of the base model's storage across multiple models. Nevertheless, this method ignores the distinction in delta weight values across different input and output channels, resulting in substantial quantization error. To mitigate this, output channel-wise quantization (Xiao et al., 2023; Wei et al., 2023; Liu et al., 2024d) where each output channel is allocated its own learnable step size, still neglects input channel variations, as shown in Figure 1. To further mitigate information loss, rescaling input channels before quantization can be employed, but this provides only limited alleviation at extremely low bitwidths.

In this paper, we propose ME-Switch, a memory-efficient expert switching framework tailored for LLMs. To reduce the quantization error while simultaneously reducing storage needs, we develop a mixed-precision quantization method that quantizes non-salient input channels of delta weights to extremely low bits while preserving those salient ones, which could substantially increase quantization errors when quantized at very low bitwidths, in full precision. Since the number of salient input channels is relatively small, incorporating a limited amount of high-precision delta weights incurs negligible memory overhead and inference cost. To identify the important input channels of delta weights, one may select based on their magnitudes (Dettmers et al., 2022), as shown in Figure 2(a), which however fails to capture those leading to high quantization error. In contrast, our approach identifies salient input channels of delta weights based on their impact on reconstruction errors in the output activations, as illustrated in Figure 2(b).

Our contributions can be summarized as follows. 1) We introduce ME-Switch, a memory-efficient framework designed for serving multiple LLMs. It only stores a single full-precision pre-trained model and dynamically loads the appropriate compressed delta model weights in response to user queries. 2) We develop a mixed-precision quantization method that significantly reduces the storage demands of serving multiple LLMs while maintaining performance, which is achieved by selectively quantizing the non-salient input channels of delta weights and leaving the salient ones unchanged. 3) We conduct extensive experiments demonstrating the promising memory efficiency and accuracy of ME-Switch. Remarkably, when serving three models from the Mistral-7B family, ME-Switch not only delivers near-lossless performance on instruction, mathematical reasoning, and code generation tasks but also reduces the model size by 2.04×. More impressively, our method is able to serves up to 16 Mistral-7B models on a single NVIDIA A100 GPU without running out of memory.

## 2 RELATED WORK

**Efficient LLM serving.** LLMs can be efficiently deployed on GPUs for high-throughput serving using several inference frameworks, such as vLLM (Kwon et al., 2023) and Orca (Yu et al., 2022). Given that a single LLM cannot excel across all domains, it is crucial to serve multiple LLMs simultaneously to handle diverse user queries effectively. To determine which model to use during inference, Zooter (Lu et al., 2023) utilizes a task-level router to load different LLMs based on the incoming task requirements. This approach, however, introduces significant memory challenges, as hosting all LLMs on a GPU simultaneously can excessively strain the available VRAM. One solution is to serve multiple Low-Rank Adaptation (LoRA) modules within a multi-tenant serving system, such as Punica (Chen et al., 2023) and S-LoRA (Sheng et al., 2023a). However, these methods fail to accommodate full fine-tuned models because such models typically do not display the low-rank characteristics necessary for LoRA approximation. Another viable strategy to address the above issue involves dynamically loading different LLMs from CPU memory to GPU memory as needed, thereby reducing peak GPU memory utilization. Nevertheless, this dynamic loading competes for GPU memory bandwidth with model-swapping operations, presenting a significant bottleneck. To mitigate these challenges, techniques like Deltazip (Yao & Klimovic, 2023) and BitDelta (Liu et al., 2024b) have been developed to compress the delta parameters. These methods allow task-specific delta parameters to be loaded into GPU memory on-demand, ensuring that only a single pre-trained model's parameters reside permanently in GPU memory. This approach aims to achieve low-latency inference while minimizing the costs associated with maintaining numerous fine-tuned models in GPU memory. Our approach stands out by serving multiple LLMs with a minimal memory footprint with nearly lossless performance, which is achieved through an advanced delta weights quantization.

**Delta compression.** In recent years, numerous approaches have focused on reducing the storage overhead for maintaining different task-specific models through delta parameter compression. Model merging strategies often incorporate multiple tasks' delta parameters (Ding et al., 2023) into the pretrained parameters to minimize the number of parameters needed for multi-task operations. To address the parameter conflicts that often arise during model merging, Yu et al. (Yu et al., 2023a) proposed a method involving massive unstructured random pruning (achieving 90% sparsity) of delta parameters and subsequent rescaling. This approach ensures that the accuracy of downstream tasks is not compromised. In a similar vein, Ties-merging (Yadav et al., 2024) developed a pruning strategy based on the magnitude and sign of delta parameters to further reduce conflicts and storage needs. Additionally, the LoRA-based PEFT methods (Hu et al., 2021; Valipour et al., 2022; Ping et al., 2024) introduce a novel approach by learning one or several low-rank matrices to represent the delta parameters. ZipLoRA (Shah et al., 2023) explores the sparsity within these low-rank matrices, allowing the fusion of low-rank matrices from different tasks to further reduce the spatial overhead of multi-task models. However, these methods struggle to accurately approximate delta parameters in full fine-tuned models, as these models usually lack low-rank properties (Liu et al., 2024b; Lialin et al., 2024; Hao et al., 2024). Conversely, some studies focus solely on compressing delta parameters for each task without merging, thereby mitigating performance degradation across multiple tasks. For example, Ryu et al. (Ryu et al., 2023) and Isik et al. (Isik et al., 2023) combine quantization and low-rank estimation techniques to reduce the storage size of delta parameters. Liu et al. (Liu et al., 2024b) push this further by quantizing each task's delta parameters to 1 bit, achieving more than a tenfold compression. However, these quantization methods often lack robust handling of outliers, resulting in performance declines compared to uncompressed delta parameters. Moreover, models that do not integrate multi-task delta parameters require manual activation of specific delta parameters for each task, which reduces the model's applicability in multi-task environments.

## 3 PRELIMINARIES

For simplicity, we employ uniform quantization (Jacob et al., 2018) to compress the models. Given a matrix $\mathbf{X}$ with floating-point values (*e.g.*, BF16), the quantization process can be expressed as:

$$\hat{\mathbf{X}} = \mathrm{quant}(\mathbf{X}) = \mathrm{clamp}\left(\lfloor \tfrac{\mathbf{X}}{s} \rceil, -Q_N, Q_P\right) \times s, \tag{1}$$

where the function $\mathrm{clamp}(\mathbf{V}, \mathbf{V}_{\min}, \mathbf{V}_{\max})$ clamps all elements in $\mathbf{V}$ within the range $[\mathbf{V}_{\min}, \mathbf{V}_{\max}]$, the operator $\lfloor \cdot \rceil$ rounds a given value to the nearest integer, and $s$ is a learnable quantization step size initialized by $\max(|\hat{\mathbf{X}}|)/(2^{b-1} - 1)$. Here, $Q_N$ and $Q_P$ denote the number of negative and

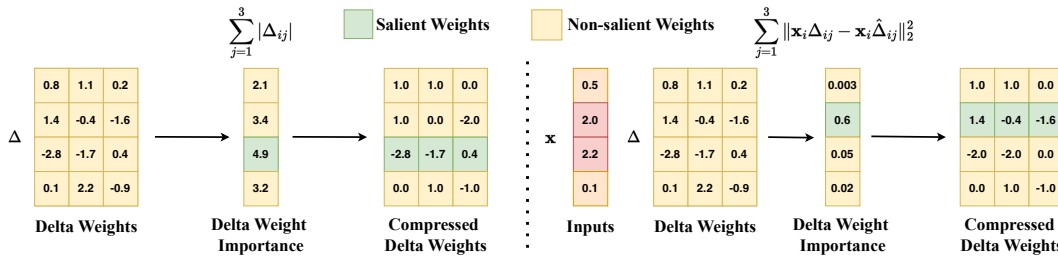

(a) Magnitude-based selection        (b) Our reconstruction-error-based selection

Figure 2: An illustration comparison between the magnitude-based selection of salient delta weights and our reconstruction-error-based selection method. Given a delta weight matrix $\Delta \in \mathbb{R}^{m \times n}$, its quantized version $\hat{\Delta}$, and input $\mathbf{x} \in \mathbb{R}^m$, where $m$ and $n$ denote the number of input and output channels, respectively, our method measures the importance of each input delta channel by $\sum_{j=1}^{n} \|\mathbf{x}_i \Delta_{ij} - \mathbf{x}_i \hat{\Delta}_{ij}\|_2^2$.

positive quantization levels, respectively. For $b$-bit quantized weights, $Q_N$ and $Q_P$ are set to $2^{b-1}$ and $2^{b-1} - 1$, respectively. Since the rounding function is not differentiable, we use the straight-through estimator (STE) (Esser et al., 2020) for gradient approximation following (Esser et al., 2020). For binary quantization, we use the quantization method following (Rastegari et al., 2016).

Recent studies (Liu et al., 2024b; Yao & Klimovic, 2023) have shown that the weights of a fine-tuned model can be decomposed into the weights of the pre-trained model and the delta weights introduced during fine-tuning. Let $\mathbf{W} \in \mathbb{R}^{m \times n}$ and $\mathbf{W}_{\text{FT}} \in \mathbb{R}^{m \times n}$ be the weight matrices of the pre-trained model and the fine-tuned model, respectively, where $m$ represents the number of input channels and $n$ denotes the output number of channels. The delta weight is defined as $\Delta = \mathbf{W}_{\text{FT}} - \mathbf{W}$. To reduce storage requirements, one can perform quantization using Eq. (1) to compress $\Delta$. However, this method often results in significant performance degradation because it assumes all delta weight channels are equally sensitive to quantization noise.

## 4 PROPOSED METHOD

In this section, we propose ME-Switch, a memory-efficient expert switching framework designed to optimize the deployment of multiple LLMs. ME-Switch reduces memory demands while maintaining performance by using mixed-precision quantization, which quantizes non-salient weights to extremely low bitwidths while keeping salient weights intact.

### 4.1 SALIENT-AWARE DELTA COMPRESSION

In this section, we introduce our salient-aware delta compression approach, which specifically targets the preservation of salient input channels during quantization. While quantization methods with learnable step sizes for each output channel can handle variations in output channels effectively (Xiao et al., 2023; Liu et al., 2024d), variations in input channels pose a significant challenge to maintaining model performance, as shown in Figure 1. Some salient input channels of delta weights, regardless of their magnitude, can cause substantial information loss and degrade model performance when quantized to a very low bitwidth. To address this, rescaling the input channels of delta weights before quantization (Lin et al., 2024) offers a solution to mitigate some of the quantization error, which however provides only limited alleviation under the context of extremely low-bit quantization.

**Salient-aware delta selection.** To protect salient input channels of delta weights, we develop a mixed-precision quantization method, where the majority of input channels of delta weights are quantized to low-bit precision, while only a small number of critical input channels are represented in their original precision. This approach achieves significant reductions in storage requirements with minimal performance loss. The remaining challenge lies in identifying these salient input channels. A naive approach might involve selecting the input channels with large magnitudes as the salient delta weights (Dettmers et al., 2022), as shown in Figure 2(a). However, this method neglects the influence of input activations on the outputs, failing to identify channels that lead to high quantization

error. To address this, inspired by the pruning metric (Sun et al., 2024), we introduce a salient delta weights selection metric based on reconstruction errors in the outputs, considering both weights and input activations, as shown in Figure 2(b). Specifically, given an input $\mathbf{x} \in \mathbb{R}^m$, the reconstruction error for the input channel $i$ can be calculated by

$$\sum_{j=1}^{n} \|\mathbf{x}_i \Delta_{ij} - \mathbf{x}_i \hat{\Delta}_{ij}\|_2^2, \tag{2}$$

where $\hat{\Delta}$ is the quantized delta weights and $n$ denotes the output channel number. With the reconstruction error defined, we choose those input channels with the top-$k$ largest reconstruction errors as the salient ones and retain them in 16-bit representation to maintain performance.

**Efficient distillation.** The quantization step size in Eq. (1) plays an important role in the final performance. To learn the quantization step size, we employ knowledge distillation to guide the alignment of the output logits of the quantized model with those of the full-precision fine-tuned model following (Liu et al., 2024b). To reduce training overhead, we freeze the delta weights and focus solely on optimizing the quantization step size $\mathbf{s}$ using a small calibration dataset $\mathcal{X}$ by solving

$$\arg\min_{\mathbf{s}} \|f(\mathcal{X}) - \hat{f}(\mathcal{X}; \mathbf{s})\|_2^2, \tag{3}$$

where $f(\cdot)$ and $\hat{f}(\cdot)$ denote the output logits of the fine-tuned and quantized models, respectively. Thanks to the reduced number of trainable parameters, this training process is highly efficient.

**On-demand swapping.** The extremely low-bit compression for delta weights significantly reduces the model size, alleviating storage demands. This approach enables us to maintain a single pre-trained model while storing multiple sets of compressed delta weights, facilitating efficient on-demand swapping. In this scenario, the pre-trained model remains in GPU memory, and the corresponding compressed delta weights are loaded dynamically based on the user query. Compared with directly serving multiple models, our method is more GPU memory-efficient. Since LLM decoding is memory-bound (Liu et al., 2023; Sheng et al., 2023b) due to the auto-regressive nature, reducing model size effectively decreases the parameter loading time, thereby improving decoding latency. To achieve fast inference, we decouple the matrix multiplication during inference into two components:

$$\mathbf{y} = \mathbf{x}\mathbf{W}_{\text{FT}} = \mathbf{x}(\mathbf{W} + \Delta) \approx \mathbf{x}\mathbf{W} + \mathbf{x}\tilde{\Delta}, \tag{4}$$

where $\tilde{\Delta}$ represents the compressed delta weights, including both the quantized unsalient and the full-precision salient delta weights. For $\mathbf{x}\mathbf{W}$, the computation is performed using a BF16 batched GEMM kernel. For $\mathbf{x}\tilde{\Delta}$, we implement an efficient Triton kernel (Tillet et al., 2019) that fuses dequantization and matrix multiplication for efficient computation, reducing intermediate memory operations and eliminating unnecessary data transfers. The reduced memory footprint of the compressed delta weights enables our method to perform batched forward passes across multiple models simultaneously. This batching at the model level significantly enhances efficiency compared to the traditional approach, which processes each model individually—especially beneficial when serving multiple models (See Figure 7). The pseudo codes of our Triton kernel can be found at Section C of the appendix.

### 4.2 MODEL SIZE REDUCTION ANALYSIS

Let $\Psi$ be the model size of an BF16 pre-trained model. When storing $M$ BF16 models, the total model size is $M\Psi$. In contrast, using our method, we store a single base model and $M$ compressed delta models. The total model size is $\Psi + M\tilde{\Psi}$, where $\tilde{\Psi}$ represents the size of the compressed delta model. Therefore, the compression ratio can be computed by $M\Psi/(\Psi + M\tilde{\Psi})$. For example, when serving 9 LLaMA-2-13B models, our method achieves a compression ratio of $3.85\times$. Empirical studies on compression ratios for varying model numbers are shown in Figures 5 and 6.

## 5 EXPERIMENTS

**Candidate LLMs.** We apply our ME-Switch to three model families, Mistral-7B (Jiang et al., 2023), LLaMA-2-13B (Touvron et al., 2023b), and LLaMA-3-8B (Dubey et al., 2024). For the Mistral family, we include Dolphin-2.2.1-Mistral-7B [3] as the instruction expert, Speechless-Code-Mistral-

---

[3]https://huggingface.co/cognitivecomputations/dolphin-2.2.1-mistral-7b

7B [4] as the code expert, and MetaMath-Mistral-7B [5] as the math expert. To show how our method generalizes, we further include BioMistral-7B [6] as an medical expert and Saul-7B-Base [7] as an legal expert. Both models are fine-tuned from Mistral-7B-Instruct-v0.1 [8]. For the LLaMA-2-13B family, LLaMA-2-13B-Chat (Touvron et al., 2023b) serves as the instruction expert, MetaMath-13B [9] as the math expert, and LLaMA2-Chinese-13B-Chat [10] as the Chinese expert. For the LLaMA-3-8B family, we include LLaMA-3-8B-Instruct [11] as the instruction expert and LLaMA-8B-Chinese-Chat [12] as the Chinese expert. The above models are fine-tuned based on pre-trained backbones. All pre-trained models used in our experiments are converted to BF16.

**Training and testing datasets.** We collect a diverse set of instruction samples from various open-source datasets, including Alpaca (Taori et al., 2023) for the instruction domain, MetaMathQA (Yu et al., 2023b) for the mathematics domain, Code-74k-ShareGPT [13] for the code domain, BioInstructQA [14] for the medical domain, LegalBench-Instruct [15] for the legal domain, and Chinese Alpaca [16] for the Chinese domain. To measure the performance of the resulting LLMs, we report accuracy on several benchmarks across different domains: MMLU (Hendrycks et al., 2021a) for the instruction, GSM8K (Cobbe et al., 2021) and MATH (Hendrycks et al., 2021b) for the mathematics, HumanEval (Chen et al., 2021) and MBPP (Austin et al., 2021) for the code, and C-Eval (Huang et al., 2024b) and C-MMLU (Li et al., 2023) for the Chinese. We use the WizardCoder toolbox to evaluate on HumanEval and MBPP, and the OpenCompass toolbox (Contributors, 2023) to evaluate on other datasets. For MMLU, we report accuracy based on 5-shot in-context learning. To determine the answer for each question, we assess the perplexity of various response options and select the one with the lowest perplexity. For GSM8K and MATH, we adopt a 4-shot Chain of Thought (CoT) methodology to obtain the final answer following (Wei et al., 2022). For the HumanEval and MBPP datasets, we employ a 0-shot configuration and generate answers using greedy decoding. We assess the functional correctness using the pass@1 metric following (Liu et al., 2024c). For the medical and legal domains, we evaluate model performance using the respective subsets of the MMLU dataset.

**Implementation details.** For salient-aware delta compression, we construct a calibration set from each domain-specific dataset and use these sets to compress the delta weights for each respective domain. Each calibration set consists of 1600 randomly sampled sequences, each with a length of 128 tokens. The bitwidth $b$ and the number of BF16 input channels $k$ are set to 2 and 8, respectively. We use the AdamW optimizer (Loshchilov & Hutter, 2019) with a learning rate of $10^{-5}$ and a mini-batch size of 4 for training over 1 epoch. Delta weights compression experiments for the Mistral-7B and LLaMA-3-8B famles are conducted on two NVIDIA A100 80G GPUs, while for the LLaMA-2-13B model, we use four NVIDIA A100 80G GPUs.

## 5.1 MAIN RESULTS

To evaluate the efficacy of our proposed model, we apply ME-Switch to the Mistral-7B, LLaMA-3-8B and LLaMA-2-13B model families. The experimental results, detailed in Tables 1, 2, 3 and 4, demonstrate that ME-Switch, even with extremely compressed delta weights, achieves performance comparable to that of the respective unquantized expert models across various downstream tasks. For the Mistral-7B family, on MMLU, ME-Switch lags behind the math expert by just 0.22% in mathematical reasoning tasks. Notably, ME-Switch consistently outperforms the code expert in code generation tasks. The performance improvements over uncompressed expert models are primarily attributed to additional training through efficient distillation, which improves the models' task-specific

---

[4] https://huggingface.co/uukuguy/speechless-code-mistral-7b-v1.0

[5] https://huggingface.co/meta-math/MetaMath-Mistral-7B

[6] https://huggingface.co/BioMistral/BioMistral-7B

[7] https://huggingface.co/Equall/Saul-7B-Base

[8] https://huggingface.co/mistralai/Mistral-7B-Instruct-v0.1

[9] https://huggingface.co/meta-math/MetaMath-13B-V1.0

[10] https://huggingface.co/FlagAlpha/Llama2-Chinese-13b-Chat

[11] https://huggingface.co/meta-llama/Meta-Llama-3-8B-Instruct

[12] https://huggingface.co/shenzhi-wang/Llama3-8B-Chinese-Chat

[13] https://huggingface.co/datasets/ajibawa-2023/Code-74k-ShareGPT

[14] https://huggingface.co/datasets/BioMistral/BioInstructQA

[15] https://huggingface.co/datasets/Equall/legalbench_instruct

[16] https://huggingface.co/datasets/hfl/alpaca_zh_51k

Table 1: Main results for Mistral-7B and LLaMA-2-13B families.

| Model | MMLU (%) ↑ | | | | | Mathematical Reasoning (%) ↑ | | | Code Generation (%) ↑ | | |
|---|---|---|---|---|---|---|---|---|---|---|---|
| | STEM | Hums. | Social | Other | Avg. | GSM8K | Math | Avg. | HumanEval | MBPP | Avg. |
| Dolphin-2.2.1-Mistral-7B | 52.05 | 68.83 | 73.42 | 65.43 | 63.43 | 63.68 | 12.80 | 38.24 | 42.70 | 54.90 | 48.80 |
| MetaMath-Mistral-7B | 50.45 | 66.82 | 71.63 | 64.60 | 61.87 | 73.92 | 20.62 | **47.27** | 0.00 | 21.60 | 10.80 |
| Speechless-Code-Mistral-7B | 51.82 | 68.35 | 73.74 | 65.69 | 63.36 | 61.18 | 13.52 | 37.35 | 51.20 | 60.40 | 55.80 |
| ME-Switch | 53.17 | 69.09 | 73.88 | 65.40 | **63.95** | 73.62 | 20.48 | 47.05 | 51.80 | 60.70 | **56.25** |
| Model | MMLU (%) ↑ | | | | | Mathematical Reasoning (%) ↑ | | | Chinese (%) ↑ | | |
| | STEM | Hums. | Social | Other | Avg. | GSM8K | Math | Avg. | C-Eval | C-MMLU | Avg. |
| LLaMA-2-13B-Chat | 44.26 | 59.79 | 63.20 | 56.57 | 54.60 | 43.75 | 5.20 | 24.48 | 36.13 | 38.71 | 37.42 |
| MetaMath-13B | 37.81 | 52.77 | 56.00 | 50.05 | 47.84 | 69.14 | 8.48 | 38.81 | 33.62 | 32.70 | 33.16 |
| LLaMA2-Chinese-13B-Chat | 45.24 | 60.01 | 62.47 | 55.92 | 54.67 | 38.89 | 4.54 | 21.72 | 40.28 | 39.16 | 39.72 |
| ME-Switch | 44.57 | 60.87 | 64.00 | 58.04 | **55.45** | 70.05 | 13.20 | **41.63** | 40.13 | 39.91 | **40.02** |

Table 2: Main results for LLaMA-3-8B family. "BF16 Baseline" refers to the performance metrics of experts without compression.

| Method | MMLU (%) ↑ | | | | | Chinese (%) ↑ | | |
|---|---|---|---|---|---|---|---|---|
| | STEM | Hums. | Social | Other | Avg. | C-Eval | C-MMLU | Avg. |
| BF16 Baseline | 57.30 | 71.64 | 77.83 | 71.13 | **68.05** | 51.99 | 52.25 | 52.12 |
| ME-Switch | 57.12 | 70.89 | 78.60 | 70.92 | 67.93 | 52.67 | 52.70 | **52.69** |

Table 3: Results on the legal domain for the Mistral-7B family.

| Method | BF16 Baseline | ME-Switch |
|---|---|---|
| International Law | 74.38 | 75.76 |
| Jurisprudence | 71.30 | 67.59 |
| Professional law | 43.02 | 43.68 |
| Avg. | **62.90** | 62.34 |

Table 4: Results on the medical domain for the Mistral-7B family.

| Method | Clinical Knowledge | Medical Genetics | Anatomy | Professional Medicine | College Biology | College Medicine | Avg. |
|---|---|---|---|---|---|---|---|
| BF16 Baseline | 64.53 | 69.00 | 57.89 | 57.72 | 58.33 | 58.38 | **60.98** |
| ME-Switch | 62.26 | 68.00 | 48.89 | 57.72 | 63.89 | 61.27 | 60.34 |

Table 5: Comparisons between fixed-precision quantization and mixed-precision quantization for MetaMath-Mistral-7B and Speechless-Code-Mistral-7B.

| Method | Model Size (GB) | Mathematical Reasoning (%) ↑ | | | Code Generation (%) ↑ | | |
|---|---|---|---|---|---|---|---|
| | | GSM8K | Math | Avg. | HumanEval | MBPP | Avg. |
| BF16 Baseline | 13.48 | 73.92 | 20.62 | 47.27 | 51.20 | 60.40 | 55.80 |
| Fixed-precision | 2.11 | 73.31 | 20.44 | 46.88 | 47.00 | 59.10 | 53.05 |
| Mixed-precision | 2.13 | 73.62 | 20.48 | **47.05** | 51.80 | 60.70 | **56.25** |

Table 6: Effect of different BF16 input channel numbers $k$ for Speechless-Code-Mistral-7B.

| Model | Model Size (GB) | HumanEval | MBPP | Avg. |
|---|---|---|---|---|
| BF16 | 13.48 | 51.20 | 60.40 | 55.80 |
| $k = 8$ | 2.13 | 51.80 | 60.70 | 56.25 |
| $k = 16$ | 2.15 | 49.40 | 61.90 | 55.65 |
| $k = 32$ | 2.19 | 51.20 | 61.20 | 56.20 |
| $k = 64$ | 2.26 | 51.20 | 61.60 | **56.40** |

performance by optimizing the quantization step size. Similar phenomena are also observed in many quantization literature (Esser et al., 2020; Yamamoto, 2021; Liu et al., 2022).

## 5.2 Ablation Studies

**Fixed-precision quantization *vs.* Mixed-precision quantization.** To validate the effect of mixed-precision quantization, we compress the delta weights of MetaMath-Mistral-7B and Speechless-Code-Mistral-7B using both fixed-precision quantization and our salient-aware mixed-precision quantization. We evaluate their performance on mathematical reasoning and code generation tasks, respectively. The bitwidth $b$ and the number of BF16 input channels $k$ are set to default values as specified in the implementation details. The results in Table 5 indicate that despite retaining a minimal number of BF16 channels, the model size of the mixed-precision model (2.13 GB) is nearly identical compared to fixed-precision model (2.11 GB). However, introducing a small number of BF16 channels significantly improves performance. For example, our compressed Speechless-Code-Mistral-7B, with a model size reduced by $6.33\times$, even outperforms the full-precision counterpart by 0.45% in the average accuracy on code generation tasks. This underscores the capability of salient-aware mixed-precision quantization to minimize model size while preserving model performance.

**Low-rank adaptation *vs.* our salient-aware delta compression.** In addition to mixed-precision quantization, we can also employ low-rank adaptation (LoRA) to compress delta weights. Specifically, we decompose the delta weights as $\Delta = \mathbf{U}\mathbf{\Sigma}\mathbf{V}$ and approximate delta weights using low-rank approximation $\tilde{\Delta} = \mathbf{A}\mathbf{B}$ where $\mathbf{A} = \tilde{\mathbf{U}}\sqrt{\tilde{\mathbf{\Sigma}}}$ and $\mathbf{B} = \sqrt{\tilde{\mathbf{\Sigma}}}\tilde{\mathbf{V}}$. Subsequently, $\mathbf{A}$ and $\mathbf{B}$ are refined using our efficient distillation mentioned in Section 4.1. To compare the effectiveness of LoRA against mixed-precision quantization, we apply both methods to the delta weights of Dolphin-2.2.1-Mistral-7B and MetaMath-Mistral-7B and evaluate performance on instructional and mathematical reasoning tasks. For LoRA, we set the rank to 512. The results are detailed in Table 7. Our approach with a much smaller model size outperforms LoRA, especially on Math. These results reveal that LoRA cannot accurately approximate delta weights for full fine-tuned models like Dolphin-2.2.1-Mistral-7B and MetaMath-Mistral-7B. To show the underlying reason, we show the cumulative energy of delta weights for MetaMath-Mistral-7B in Figure 3, using squared singular values to measure the "energy" of the projection matrix. The results show that all projection layers consistently exhibit a similar trend and possess a relatively high rank. Therefore, due to the absence of low-rank properties, LoRA cannot accurately approximate the delta weights of full fine-tuned models.

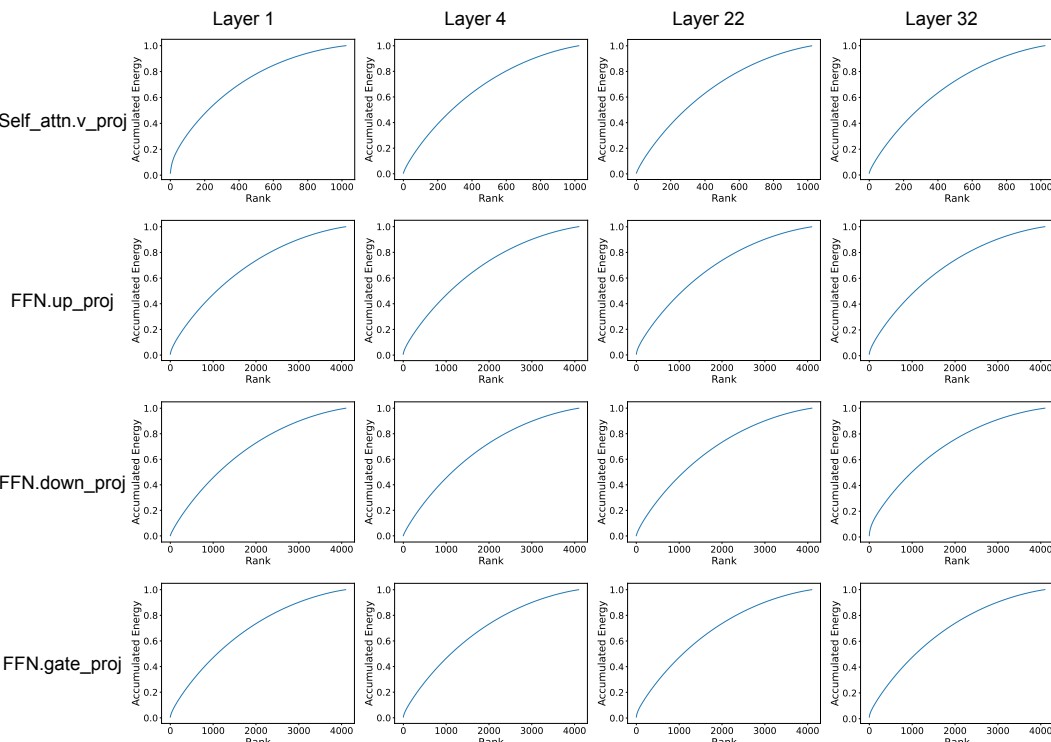

Figure 3: An illustration showing the cumulative energy of delta weights for MetaMath-Mistral-7B model derived through Singular Value Decomposition (SVD).

**Effect of different number of BF16 channels.** To assess the impact of varying BF16 input channel counts $k$, we compress the delta weights of Speechless-Code-Mistral-7B and evaluate performance on code generation tasks. The bitwidth $b$ is set at 2. From Table 6, our method already achieves lossless performance with $k = 8$. Increasing $k$ further yields no significant performance improvements, indicating that performance has plateaued. Therefore, we set $k$ to 8 by default.

**Effect of different quantization bitwidths.** To investigate the impact of varying bitwidths $b$, we compress the delta weights of Speechless-Code-Mistral-7B and evaluate the performance on a code generation task. Table 8 shows that increasing $b$ from 1 to 2 significantly improve performance, achieving lossless results. Given that further increasing $b$ lead to negligible performance differences due to saturation, we set $b$ to 2 as the default.

**Performance comparisons with other weight-only quantization methods.** To demonstrate the promising performance of our salient-aware delta compression, we include the following weight-only quantization methods: **AWQ**: we use AWQ (Lin et al., 2024) to rescale input channels of delta

Table 7: Performance comparisons between LoRA and our salient-aware delta compression for Dolphin-2.2.1-Mistral-7B and MetaMath-Mistral-7B.

Table 8: Effect of different bitwidths $b$ for Speechless-Code-Mistral-7B.

| Model | Model Size (GB) | MMLU (%) ↑ | | | | | Mathematical Reasoning (%) ↑ | | |
|---|---|---|---|---|---|---|---|---|---|
| | | STEM | Hums. | Social | Other | Avg. | GSM8K | Math | Avg. |
| LoRA | 2.99 | 52.16 | 68.38 | 73.15 | 65.52 | 63.33 | 73.62 | 17.30 | 45.46 |
| Ours | **2.11** | 53.17 | 69.09 | 73.88 | 65.40 | **63.95** | 73.62 | 20.48 | **47.05** |

| Model | HumanEval | MBPP | Avg. |
|---|---|---|---|
| BF16 | 51.20 | 60.40 | 55.80 |
| $b = 1$ | 49.40 | 49.90 | 49.65 |
| $b = 2$ | 51.80 | 60.70 | **56.25** |
| $b = 4$ | 51.80 | 60.20 | 56.00 |

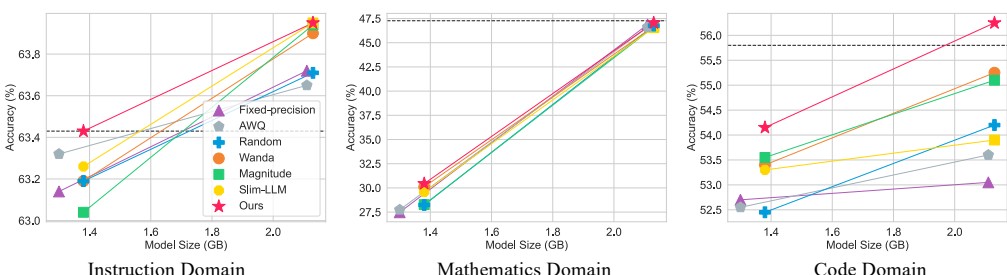

Figure 4: Average accuracy *vs.* delta weights size across different domains. "Baseline" refers to the fixed-precision quantization baseline. The dashed line indicates the full-precision counterpart.

weights before quantization to mitigate quantization errors. **Random**: using our salient-aware delta compression, we randomly select some input channels from the delta weights as important channels. **Wanda**: leveraging our salient-aware delta compression, we select important input channels of delta weights using the pruning metric from Wanda (Sun et al., 2024). **Magnitude**: within our salient-aware delta compression framework, we select sensitive input channels of delta weights based on their weight magnitude, following the method proposed by (Dettmers et al., 2022). **Slim-LLM**: using the saliency metric in Slim-LLM (Huang et al., 2024a) to select the important channels. We also include fixed-precision quantization for comparisons. We applied all methods to compress the delta weights of Dolphin-2.2.1-Mistral-7B, MetaMath-Mistral-7B, and Speechless-Code-Mistral-7B, using bitwidths $b = 1$ and $b = 2$. The results are shown in Figure 4. The detailed number of different methods can be found at Section E of the appendix. From the results, we observe that AWQ achieves comparable performance to the 1-bit baseline on code domain, highlighting the limitations of rescaling in extremely low-bitwidth quantization. In contrast, keeping the salient delta input channels performs favourably against the rescaling input channel counterpart. Moreover, our salient channel selection demonstrates superior performance than Random, Wanda, Slim-LLM and Magnitude metrics across various bitwidths and tasks. For example, for $b = 2$, our salient-aware delta compression outperforms Wanda by 1.0% on the average accuracy on code domain, underscoring the effectiveness and superiority of our approach in selecting the salient delta weights.

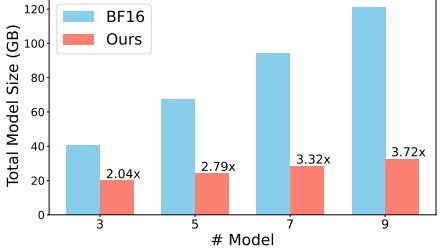

Figure 5: Model size reduction results in terms of Mistral-7B family. The model sizes for the a single 16-bit floating-point model and a compressed model are 13.48 GB and 2.13 GB, respectively.

Figure 6: Model size reduction results in terms of LLaMA-2-13B family. The model sizes for a single BF16 model and a compressed model are 24.23 GB and 3.60 GB, respectively.

**Model size reduction analysis.** To investigate the model size reduction as discussed in Section 4.2, we compare the total storage requirements of full-precision models with those of our compressed models for the Mistral-7B and LLaMA-2-13B families across varying model counts, as shown in

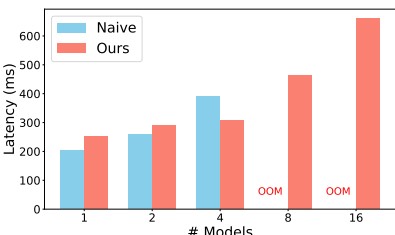

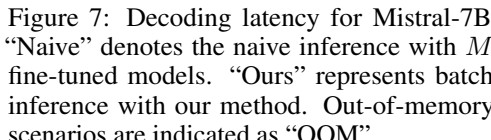

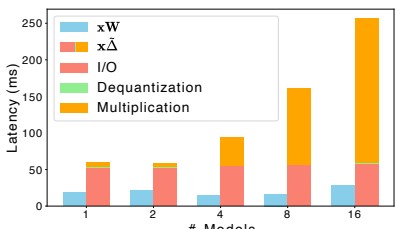

Figure 7: Decoding latency for Mistral-7B. "Naive" denotes the naive inference with $M$ fine-tuned models. "Ours" represents batch inference with our method. Out-of-memory scenarios are indicated as "OOM".

Figure 8: Latency decomposition of our method for Mistral-7B on a single NVIDIA A100 80G GPU. We show the latency of $\mathbf{xW}$ and $\mathbf{x\hat{\Delta}}$ (I/O, dequantization, and multiplication).

Figures 5 and 6. As the number of models increases, the compression ratios improve substantially. For instance, with nine models, our method achieves a $3.72\times$ reduction compared to full-precision models for Mistral-7B family. These savings become even more pronounced with larger model sizes, reaching up to a $3.85\times$ reduction for LLaMA-2-13B family.

**Latency analysis.** To assess the latency improvements from delta weights compression, we measured the end-to-end decoding latency of the Mistral-7B model with an input sequence length of 128 on a single NVIDIA A100. Decoding latency is critical, as it typically dominates processing time in LLM operations (Lin et al., 2024; Liu et al., 2023). Our efficient Triton kernel, which enables batched matrix multiplication between multiple compressed weight matrices and high-precision input activations, is compared against the conventional approach of individually processing multiple models. Results depicted in Figure 7 illustrate that while our method may perform slightly slower than the naive approach for a small number of models due to additional dequantization overhead, it provides lower latency as the number of models is greater than 4. These results show that our method scales more efficiently than the naive inference method. In the naive method, each $\mathbf{xW}$ is computed independently during the forward pass, requiring a distinct $\mathbf{W}$ for each user in the batch. As the number of models grows ($>= 4$), this approach results in substantial I/O costs due to loading of large weight matrices. In contrast, our method leverages shared pre-trained model weights $\mathbf{W}$ along with a set of small deltas $\hat{\Delta}$, significantly reducing the inference I/O burden. More importantly, unlike the naive approach, our method is able to simultaneously serve 16 models on GPUs without running into out-of-memory (OOM) issues, demonstrating better scalability and efficiency in high-load scenarios.

We further provide a detailed breakdown of decoding times for Mistral-7B model in Figure 8. We observe that the dequantization cost is very small across different model numbers. Initially, latency is dominated by I/O operations because LLM decoding is a memory-bound process when the batch size is small (Lin et al., 2024; Liu et al., 2023). However, as the number of models grows, compute-related operations, such as matrix multiplications, begin to dominate the overall latency. Notably, as the number of models increases, the increased latency attributed to $\mathbf{x\hat{\Delta}}$ exceeds that of $\mathbf{xW}$, primarily due to the increased multiplication cost of the more compressed delta weights.

## 6    CONCLUSION AND FUTURE WORK

In this paper, we have introduced ME-Switch, a memory-efficient expert switching framework designed for LLMs. Our method has addressed the critical challenge of balancing model performance with storage efficiency. The core of our ME-Switch lies in a novel mixed-precision quantization method that selectively compresses non-salient delta weights to extremely low-bit precision while preserving salient delta weights. Extensive experiments on Mistral-7B, LLaMA-2-13B, and LLaMA-3-8B families have demonstrated that ME-Switch achieves performance comparable to unquantized expert models across various tasks while significantly reducing model size. In terms of limitations, quantizing the base model itself could further reduce the overall model size. This approach would require careful consideration of the combined effects of quantizing both the base model and the delta weights to ensure performance is maintained. Furthermore, reducing the bitwidth of the KV Cache could accelerate the decoding speed, offering additional efficiency improvements.

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

# Appendix

## A  MODEL-LEVEL ROUTING

In public-facing applications or open-ended systems, user inputs may vary widely in content and intent, often lacking clear contextual information. This makes it particularly challenging to determine the appropriate model for queries in advance. Therefore, we explore a simple yet effective approach as a possible solution to determine the appropriate model for a given user query. Consider a set of LLMs represented as $\mathcal{F} = \{f_1, f_2, \cdots, f_M\}$, where $M$ denotes the number of models. Given a user query $q$, we aim to find the most suitable LLM by solving the following problem $\arg\max_{f \in \mathcal{F}} P(q, f(q))$, where $P$ is a function that measures the quality or performance of the LLM response. To simplify the routing process, we assume that each LLM in $\mathcal{F}$ specializes in distinct and non-overlapping domains such as code generation or mathematical problem solving. This setup allows us to treat the routing challenge as a multiple-choice question-answering task, where each option corresponds to a specific domain, thereby transforming the problem into a domain classification problem. For the challenging problem of overlapping domains, extending the current setup to a multi-label classification framework would be necessary, and we consider this a promising direction for future work. Note that dialogue LLMs like Qwen1.5-1.8B-Chat (Bai et al., 2023) exhibit capabilities in following instructions, which inspires us to utilize a small pre-trained LLM as a model-level router. As illustrated in Figure A, we first prompt the router with the user's question using a template designed to elicit domain classification. Specifically, when a user query is received, it is embedded into the prompt template to form a structured question, as shown in Table A. This structured question is then processed by the router to perform domain classification. Based on the router's response, we then dynamically load the corresponding compressed delta weights for the selected domain-specific model, such as a mathematical model, to generate outputs.

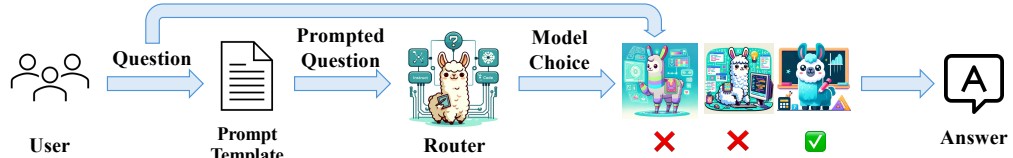

Figure A: An illustration of the model-level routing. We first prompt the model-level router with the user query using a template (See Table A for more details) that presents a list of potential domains. The router then assesses these options and selects the most relevant domain by answering a multiple-choice question, effectively classifying the query into the corresponding category.

Table A: Prompt template for model-level routing.

| PROMPT TEMPLATE FOR MODEL-LEVEL ROUTING |
| --- |
| Classify the query based on the required expertise. Route the query to the appropriate model for a precise response. Only output the letter corresponding to the best category (A, B, C, . . . , F). |
| Query: {Insert the user's query here. } |
| Options: A) Instruct - For general guidance, explanations, or broad advice. B) Code - For programming-related queries, like debugging or coding. C) Math - For mathematical inquiries, such as problems or theories. D) Chinese Language Expert - For inquiries related to the Chinese language, including translation, grammar, and usage. . . . F) {Specify additional categories and their descriptions here.} |
| Response should be only 'A', 'B', 'C', . . . or 'F', with no additional text. |

Since the router is not explicitly trained for query domain classification, its initial routing performance may be suboptimal. To improve the routing performance, we construct a multiple-choice question-answering dataset tailored for our routing problem. We collect instruction-following data from various domains and insert the query into the prompt template as shown in Figure A. The responses are constructed by considering the correct domain-specific model choice. We then fine-tune the router with our constructed dataset using supervised fine-tuning to further improve the routing accuracy.

**Comparisons with routing in Mixture of Experts (MoE).** Besides model-level routing, another approach to handle diverse user queries efficiently is to construct a MoE using a set of pre-trained

expert models. This can be achieved by integrating the feedforward layers from all pre-trained LLMs into a single MoE module at each attention-FFN block, and merging other layers, such as self-attention layers, by simply averaging their weights (Sukhbaatar et al., 2024). An additional gate network is introduced for each MoE module to perform token-level routing. However, this approach requires extensive fine-tuning of the entire network parameters and the gating network, as there is a significant gap between MoE experts and pre-trained LLMs. For example, training an MoE using existing Math, Code, and Wikipedia experts requires over 900 GPU days (Sukhbaatar et al., 2024). Notably, each expert in an MoE tends to become a generalist across all domains due to the load balancing loss, which encourages an even distribution of the workload among experts (Fedus et al., 2022; Jiang et al., 2024). This contrasts with pre-trained expert models, which are typically specialized for specific domains. Compared with MoE, our model-level routing has a significantly lower training cost, as we only need to train a router while keeping the expert models frozen.

## B    IMPLEMENTATION DETAILS OF MODEL-LEVEL ROUTING

We use Qwen1.5-1.8B-Chat (Bai et al., 2023) as the model-level router. We construct the training data using samples from various domains, as mentioned in Section A. To balance the dataset, we extract an equal number of samples from each domain-specific dataset. This constructed dataset is used to fine-tune model-level router through supervised fine-tuning for 4 epochs on a machine with 8 × A100 GPUs. We use the AdamW optimizer with $\beta_1 = 0.9$ and $\beta_2 = 0.95$, setting the learning rate to $3 \times 10^{-4}$ and applying a linear learning rate warmup. The weight decay is set to 0.01. We set the per-device mini-batch size to 8 and use gradient accumulation steps of 2.

## C    PSEUDO-CODES OF OUR TRITON KERNEL

We show the core PyTorch style pseudo-codes of the Triton kernel in Figure B.

## D    RESULTS OF MODEL-LEVEL ROUTING

We present the results of model-level routing in Table B. The results indicate that model-level routing achieves nearly lossless performance when applied to both BF16 models and our ME-Switch compressed models. For instance, for Mistral-7B family, combining ME-Switch with model-level routing achieves lossless performance on the Code domain and results in only a 0.2% accuracy drop in the Mathematical domain. These results demonstrate the effectiveness of model-level routing in accurately handling user queries.

Table B: Routing results for Mistral-7B and LLaMA-2-13B families.

| Model | MMLU (%) ↑ | | | | | Mathematical Reasoning (%) ↑ | | | Code Generation (%) ↑ | | |
|---|---|---|---|---|---|---|---|---|---|---|---|
| | STEM | Hums. | Social | Other | Avg. | GSM8K | Math | Avg. | HumanEval | MBPP | Avg. |
| Dolphin-2.2.1-Mistral-7B | 52.05 | 68.83 | 73.42 | 65.43 | 63.43 | 63.68 | 12.80 | 38.24 | 42.70 | 54.90 | 48.80 |
| MetaMath-Mistral-7B | 50.45 | 66.82 | 71.63 | 64.60 | 61.87 | 73.92 | 20.62 | **47.27** | 0.00 | 21.60 | 10.80 |
| Speechless-Code-Mistral-7B | 51.82 | 68.35 | 73.74 | 65.69 | 63.36 | 61.18 | 13.52 | 37.35 | 51.20 | 60.40 | 55.80 |
| BF16 Baseline w/ Router | 52.05 | 68.83 | 73.42 | 65.43 | **63.43** | 74.15 | 20.72 | **47.44** | 51.20 | 60.40 | 55.80 |
| ME-Switch | 53.17 | 69.09 | 73.88 | 65.40 | **63.95** | 73.62 | 20.48 | 47.05 | 51.80 | 60.70 | **56.25** |
| ME-Switch w/ Router | 51.49 | 68.37 | 73.60 | 66.08 | 63.32 | 73.39 | 20.30 | 46.85 | 51.80 | 60.70 | **56.25** |

| Model | MMLU (%) ↑ | | | | | Mathematical Reasoning (%) ↑ | | | Chinese (%) ↑ | | |
|---|---|---|---|---|---|---|---|---|---|---|---|
| | STEM | Hums. | Social | Other | Avg. | GSM8K | Math | Avg. | C-Eval | C-MMLU | Avg. |
| LLaMA-2-13B-Chat | 44.26 | 59.79 | 63.20 | 56.57 | 54.60 | 43.75 | 5.20 | 24.48 | 36.13 | 38.71 | 37.42 |
| MetaMath-13B | 37.81 | 52.77 | 56.00 | 50.05 | 47.84 | 69.14 | 8.48 | 38.81 | 33.62 | 32.70 | 33.16 |
| LLaMA2-Chinese-13B-Chat | 45.24 | 60.01 | 62.47 | 55.92 | 54.67 | 38.89 | 4.54 | 21.72 | 40.28 | 39.16 | 39.72 |
| BF16 Baseline w/ Router | 44.17 | 59.73 | 63.20 | 56.57 | 54.55 | 68.61 | 8.52 | 38.57 | 40.28 | 39.16 | 39.72 |
| ME-Switch | 44.57 | 60.87 | 64.00 | 58.04 | **55.45** | 70.05 | 13.20 | **41.63** | 40.13 | 39.91 | **40.02** |
| ME-Switch w/ Router | 44.51 | 60.87 | 64.00 | 58.04 | **55.43** | 69.90 | 13.14 | **41.52** | 40.13 | 39.84 | **39.99** |

```
864
865  def twobit_dequant_bmm_scale_kernel(a_ptr, b_ptr, c_ptr, scales_ptr, M, N, K, stride_am, stride_ak, stride_bk, stride_bn,
866      stride_cm, stride_cn, stride_scales, stride_batch_a, stride_batch_b, stride_batch_c, stride_batch_scale, BLOCK_SIZE_M: tl.
         constexpr, BLOCK_SIZE_N: tl.constexpr, BLOCK_SIZE_K: tl.constexpr, GROUP_SIZE_M: tl.constexpr, ACTIVATION: tl.constexpr,
     ):
867      """Kernel for computing the matmul C = A x B.
868      A has shape (B, M, K), float
         B has shape (B, K//n_bits, N), int, packed boolean
869      C has shape (B, M, N),
         scales is of shape (N) float16
870      """
         # -----------------------------------------------------------
871      # Map program ids 'pid' to the block of C it should compute. This is done in a grouped ordering to promote L2 data reuse.
         #       See above 'L2 Cache Optimizations' section for details.
872      pid = tl.program_id(axis=0)
         pid_batch = tl.program_id(axis=1)
873
874      num_pid_m = tl.cdiv(M, BLOCK_SIZE_M)
         num_pid_n = tl.cdiv(N, BLOCK_SIZE_N)
875      num_pid_k = tl.cdiv(K, BLOCK_SIZE_K)
876      num_pid_in_group = GROUP_SIZE_M * num_pid_n
         group_id = pid // num_pid_in_group
877      first_pid_m = group_id * GROUP_SIZE_M
         group_size_m = min(num_pid_m - first_pid_m, GROUP_SIZE_M)
878
879      pid_m = first_pid_m + (pid % group_size_m)
         pid_n = (pid % num_pid_in_group) // group_size_m
880
881      offs_m = (pid_m * BLOCK_SIZE_M + tl.arange(0, BLOCK_SIZE_M)) % M
         offs_n = (pid_n * BLOCK_SIZE_N + tl.arange(0, BLOCK_SIZE_N)) % N
882
883      offs_am = tl.max_contiguous(tl.multiple_of(offs_m, BLOCK_SIZE_M), BLOCK_SIZE_M)
         offs_bn = tl.max_contiguous(tl.multiple_of(offs_n, BLOCK_SIZE_N), BLOCK_SIZE_N)
         offs_k = tl.arange(0, BLOCK_SIZE_K)
884
885      a_ptrs = a_ptr + (offs_am[:, None] * stride_am + offs_k[None, :] * stride_ak) + pid_batch * stride_batch_a
886      # Adapted from GPTQ-Triton (https://github.com/fpgaminer/GPTQ-triton)
         # b_ptrs is set up such that it repeats elements along the K axis n_bits times
887      b_ptrs = b_ptr + ((offs_k[:, None] // 16) * stride_bk + offs_bn[None, :] * stride_bn) + pid_batch * stride_batch_b
         scales_ptrs = scales_ptr + offs_bn * stride_scales + pid_batch * stride_batch_scale
888
889      # (BLOCK_SIZE_K, BLOCK_SIZE_N)
         # shifter is used to extract each bit of each element in the int matrix
890      shifter = (offs_k % 16) * 2
         scales = tl.load(scales_ptrs)
891
892      # -----------------------------------------------------------
         # Iterate to compute a block of the C matrix.
893      # We accumulate into a '[BLOCK_SIZE_M, BLOCK_SIZE_N]' block
         # of bf32 values for higher accuracy.
894      # 'accumulator' will be converted back to bf16 after the loop.
         accumulator = tl.zeros((BLOCK_SIZE_M, BLOCK_SIZE_N), dtype=tl.float32)
895      for k in range(0, num_pid_k):
             # Load the next block of A and B, generate a mask by checking the K dimension.
896          # If it is out of bounds, set it to 0.
             a = tl.load(a_ptrs)
897          # b = tl.load(b_ptrs, mask=offs_k[:, None] < K - k * BLOCK_SIZE_K, other=0)
             b = tl.load(b_ptrs) # (BLOCK_SIZE_N,)
898
899          # Convert B from int to a.dtype
             # b: (BLOCK_SIZE_K, BLOCK_SIZE_N)
900          b = (b >> shifter[:, None]) & 0x3
             b = (b - 2).to(a.dtype)
901          b = b * scales[None, :] # BF16
             # b = b.to(a.dtype)
902
903          # We accumulate along the K dimension.
             accumulator += tl.dot(a, b)
904          # Advance the ptrs to the next K block.
             a_ptrs += BLOCK_SIZE_K * stride_ak
905          # b_ptrs += BLOCK_SIZE_K * stride_bk
             b_ptrs += (BLOCK_SIZE_K // 16) * stride_bk
906      # You can fuse arbitrary activation functions here
         # while the accumulator is still in BF32!
907      # if ACTIVATION == "leaky_relu":
         # accumulator = leaky_relu(accumulator)
908      c = accumulator.to(tl.float16)
909      # -----------------------------------------------------------
910      # Write back the block of the output matrix C with masks.
         offs_cm = pid_m * BLOCK_SIZE_M + tl.arange(0, BLOCK_SIZE_M)
911      offs_cn = pid_n * BLOCK_SIZE_N + tl.arange(0, BLOCK_SIZE_N)
         c_ptrs = (
912          c_ptr
             + stride_cm * offs_cm[:, None]
913          + stride_cn * offs_cn[None, :]
             + pid_batch * stride_batch_c
914      )
         c_mask = (offs_cm[:, None] < M) & (offs_cn[None, :] < N)
915      tl.store(c_ptrs, c, mask=c_mask)
916
917
```

Figure B: PyTorch style pseudo codes of channel disassembly and assembly during runtime.

## E    MORE PERFORMANCE COMPARISONS WITH DIFFERENT WEIGHT-ONLY QUANTIZATION METHODS

We present the detailed results of Figure 4 in Table C. A comprehensive analysis is available in Section 5.2.

Table C: Performance comparisons with different weight-only quantization methods.

| Domain | Dataset | BF16 | 1-bit | AWQ | Random | Wanda | Slim-LLM | Magnitude | Ours |
|---|---|---|---|---|---|---|---|---|---|
| Instruct (%) ↑ | MMLU | 63.43 | 63.14 | 63.32 | 63.19 | 63.19 | 63.26 | 63.04 | **63.43** |
| | GSM8K | 73.92 | 53.45 | 53.75 | 54.66 | 58.07 | 57.16 | 54.89 | 59.14 |
| Math (%) ↑ | Math | 20.62 | 1.50 | 1.72 | 1.82 | 2.14 | 2.00 | 1.64 | 1.74 |
| | Avg. | 47.27 | 27.48 | 27.74 | 28.24 | 30.11 | 29.58 | 28.27 | **30.44** |
| | HumanEval | 51.20 | 47.00 | 47.00 | 46.30 | 48.20 | 48.20 | 48.20 | 47.60 |
| Code (%) ↑ | MBPP | 60.40 | 58.40 | 58.10 | 58.60 | 58.60 | 58.40 | 58.90 | 60.70 |
| | Avg. | 55.80 | 52.70 | 52.55 | 52.45 | 53.40 | 53.30 | 53.55 | **54.15** |
| Domain | Dataset | BF16 | 2-bit | AWQ | Random | Wanda | Slim-LLM | Magnitude | Ours |
| Instruct (%) ↑ | MMLU | 63.43 | 63.72 | 63.65 | 63.71 | 63.90 | 63.68 | 63.94 | **63.95** |
| | GSM8K | 73.92 | 73.31 | 73.24 | 73.01 | 72.71 | 72.93 | 73.16 | 73.62 |
| Math (%) ↑ | Math | 20.62 | 20.44 | 19.98 | 20.52 | 20.64 | 20.16 | 20.08 | 20.48 |
| | Avg. | 47.27 | 46.88 | 46.61 | 46.77 | 46.68 | 46.55 | 46.62 | **47.05** |
| | HumanEval | 51.20 | 47.00 | 47.00 | 48.80 | 50.60 | 48.20 | 50.00 | 51.80 |
| Code (%) ↑ | MBPP | 60.40 | 59.10 | 60.20 | 59.60 | 59.90 | 59.60 | 60.20 | 60.70 |
| | Avg. | 55.80 | 53.05 | 53.60 | 54.20 | 55.25 | 53.90 | 55.10 | **56.25** |

## F    EFFECT OF SUPERVISED FINE-TUNING IN MODEL-LEVEL ROUTING

To investigate the effect of supervised fine-tuning (SFT) on model-level routing, we evaluate the domain classification performance of the router (*i.e.*, Qwen1.5-1.8B-Chat) across four domains: instruction, mathematics, code, and Chinese. As shown in Figure C, the pre-trained router performs poorly in domain classification without fine-tuning, achieving a Top-1 accuracy of only 5.80% on C-Eval. However, with SFT, the router's performance improves significantly, reaching nearly 100% accuracy across all domains. This demonstrates that supervised fine-tuning greatly enhances the instruction-following capabilities of the router, thereby improving its routing performance.

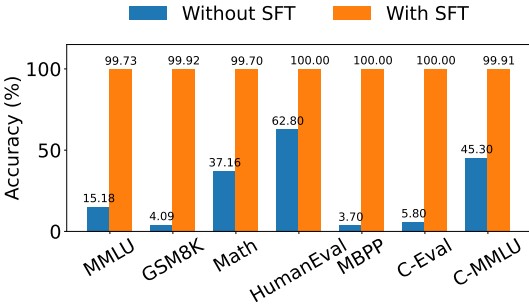

Figure C: Effect of supervised fine-tuning (SFT) in model-level routing. We assess the performance of routing by measuring the accuracy on a 4-domain classification task (instruction, mathematics, code, and Chinese).

## G    BERT *vs.* SMALL LLM FOR MODEL-LEVEL ROUTING

In addition to smaller LLMs, we can also use BERT for domain classification given user queries. Specifically, we employ DistillBERT (Sanh, 2019) as the backbone and fine-tune it on our collected dataset (described in Section A) with just the queries and corresponding domain labels. We show the router accuracy in Table D and the latency comparisons in Table E. From the results, DistillBERT

with a faster response time performs well across most datasets, except for MMLU where its limited capacity struggles with complex data. In contrast, our method consistently achieves good performance across all datasets, demonstrating its effectiveness even in complex scenarios. Moreover, our router's inference latency is just 17.60ms for sequence lengths of 128, which is less than 7% of the inference time for expert models. Therefore, we continue to leverage Qwen1.5-1.8B-Chat for its proven effectiveness in challenging scenarios.

Table D: Router performance comparisons.

| Model | MMLU | GSM8K | Math | HumanEval | MBPP | C-Eval | C-MMLU |
|-------|------|-------|------|-----------|------|--------|--------|
| DistillBERT | 73.29 | **100.00** | **99.80** | 96.34 | **100.00** | 99.97 | **100.00** |
| Qwen1.5-1.8B-Chat | **99.73** | 99.92 | 99.70 | **100.00** | **100.00** | **100.00** | 99.91 |

Table E: Router latency (ms) comparisons.

| Sequence Length | 128 | 256 | 512 |
|-----------------|-----|-----|-----|
| DistillBERT | **3.70** | **3.80** | **4.90** |
| Qwen1.5-1.8B-Chat | 17.60 | 18.30 | 19.10 |

