# OpenReview forum: "ME-Switch: A Memory-Efficient Expert Switching Framework for Large Language Models"
_ICLR.cc/2025/Conference — Submitted to ICLR 2025_

### Official Review · Reviewer_Ldho · 2024-10-23

**Soundness:** 3
**Presentation:** 4
**Contribution:** 3
**Rating:** 6
**Confidence:** 4

**Summary:**

The author contributed a memory-efficient expert-switching framework for LLMs, called ME-Switch. The framework is designed to address the challenge of LLMs' storage efficiency when serving multiple large expert models, while maintaining performance through model-level router fine-tuning and efficient distillation fine-tuning.

The proposed method applies mixed-precision quantization for delta weights to preserve model performance and enhance efficiency. This is achieved by selecting non-salient input channels based on quantization errors. Additionally, the approach incorporates a model-level router, implemented through supervised fine-tuning of a relatively small LLM, to switch between expert domain-quantized LLMs according to the user's inquiry. A detailed ablation study is included to further validate the approach.

**Strengths:**

- The author proposes an innovative and effective framework for improving the storage efficiency of serving multiple LLM experts.

    - This framework leverages mixed-precision quantization for delta weights, intelligently selecting non-salient channels based on quantization errors. The strategy of maintaining performance by selecting the top-k salient channels as non-quantized has been proven effective, as the accuracy surpasses baseline models across multiple domains.

    - The framework also introduces model-level routing by fine-tuning a small LLM router on domain-specific datasets, transforming the routing problem into a simpler domain classification task. This simplifies the problem and making the finetuning of small LLM router more lightweight compared to finetuning the MoE models

    - The framework implements on-demand swapping, enhancing GPU memory efficiency by loading only the quantized delta weights onto the GPU. This optimization is particularly effective for switching between large LLM experts. Combined with model-level routing and a specialized routing kernel, this approach significantly improves decoding efficiency. As demonstrated in the paper, it enables hosting up to 16 models on a single GPU, whereas the naive approach encounters out-of-memory (OOM) issues after just 4 models.


- A detailed ablation study is presented to demonstrate the effectiveness of mixed-precision quantization compared to baseline approaches across multiple domains. These experiments significantly strengthen the framework's methods. For example, the approach preserves accuracy competitively when compared to fixed-precision and low-rank adaptation methods. Furthermore, the ablation study clearly illustrates the selection of parameters, such as the number of non-quantized channels (k) and quantization bits (b). Lastly, the performance comparison with other weight quantization methods like AWQ, Random, Wanda, and Magnitude supports the intuition that selecting non-salient input channels yields the best performance.


- The clarity of writing, including the structure, figures and tables, is good. The author explains the background context and methodology in a good manner.

- The paper also discusses other important aspects, such as latency, with a comparison to MoE.
    - Model-level routing offers the advantage of requiring less intensive training compared to MoE.
    - The author employs a Triton kernel to further reduce latency.

**Weaknesses:**

- The model-level router relies heavily on domain classification, which requires fine-tuning the router model on domain-specific datasets. This approach may have several disadvantages:
    - The author conducted research on at most four distinct domains for the router: mathematics, code, Chinese, and instruction. Although the router demonstrates strong classification ability for these domains, it would be worthwhile to explore its performance across more domains, particularly those with potential overlap. For example, Text Classification Experts and Sentiment Analysis Experts, or Mathematics and Physics Experts, Computer Science Experts and Data Science Experts. In such scenarios, MoE could leverage the combined knowledge of multiple experts, as it is fine-tuned based on a token-level router. A potential consequence is that, when adding a large number of experts to the MoE, fine-tuning may converge faster due to shared knowledge and parameters across domain experts. In contrast, a model-level router may struggle to distinguish between overlapping domain experts and always rely on a single expert, potentially causing overall performance degradation.
    - This approach requires the construction of a domain classification dataset, which can be inconvenient. For instance, when adding a new domain expert, the router would need to be re-trained on an updated domain dataset.
    - While the approach demonstrates effectiveness compared to other quantization methods in ablation studies, the experiment results do not include a comparison with an end-to-end token-level router approach, such as MoE baselines.

- Latency analysis: The comparison only covers up to 4 models, after which the naive implementation encounters out-of-memory (OOM) issues. It is a little subtle to conclude that latency is lower than naive model as the number of models is greater than 4.

**Questions:**

- The author highlights the advantages of ME-Switch over MoE, such as reduced fine-tuning work and higher efficiency. However, it would be beneficial to include a more in-depth discussion of its limitations compared to MoE models. This would provide readers with a deeper understanding of the nature of model-level routing, particularly in addressing challenges like handling overlapping expert domains and managing a larger number of models when adapting to new experts, areas where MoE models may offer distinct advantages. Notably, the latency advantage of ME-Switch is fully realized when the number of models increases.

- Latency analysis: The discussion of latency mentions that the naive implementation has a slight advantage when the number of models is fewer than 4, as shown in Figure 8, and the author attributes this to the quantization overhead. However, why does the quantization overhead stop being the dominant factor as the number of models increases? What is the trend in the profiling of quantization overhead as the number of models continues to increase? A deeper analysis of this would be valuable.

---

> ### Author Response · Authors · 2024-11-20
> **Response to Reviewer Ldho (Part 1)**
>
> Thanks for your valuable comments.
>
> **Q1. Although the router demonstrates strong classification ability for these domains, it would be worthwhile to explore its performance across more domains, particularly those with potential overlap. In such scenarios, MoE could leverage the combined knowledge of multiple experts, as it is fine-tuned based on a token-level router. A potential consequence is that, when adding a large number of experts to the MoE, fine-tuning may converge faster due to shared knowledge and parameters across domain experts. In contrast, a model-level router may struggle to distinguish between overlapping domain experts and always rely on a single expert, potentially causing overall performance degradation.**
>
> **A1.** We acknowledge that overlapping domains can present challenges for a model-level router. However, as mentioned in Q1 of the general response, our primary contribution lies in addressing the critical challenge of **high storage demands when serving multiple LLMs**. The model-level router is presented as a **simple exploration** on automating the switching process. Handling overlapping domains would require extending the current setup to a multi-label classification framework, which we consider a promising direction for future work.
>
> Furthermore, as discussed in L295-306, our method offers a cost-efficient solution for handling diverse user queries by requiring training only the model-level router given a set of well-trained expert models. In contrast, MoE models necessitate training not just the token-level router but also all network parameters. For example, training an MoE using existing Math, Code, and Wikipedia experts requires over 900 GPU days (as reported by Sukhbaatar et al., 2024), whereas our approach only requires training at the hours level, completing within a single GPU day, making it significantly more efficient. This efficiency difference stems from the distinct characteristics of experts in MoE models versus existing well-trained experts. In MoE models, each expert often acts as a generalist across all domains due to the constraints imposed by the load balancing loss (Fedus et al., 2022; Jiang et al., 2024). In contrast, existing well-trained experts typically specialize in specific domains, which align more naturally with our approach.
>
> **Q2. This approach requires the construction of a domain classification dataset, which can be inconvenient. For instance, when adding a new domain expert, the router would need to be re-trained on an updated domain dataset.**
>
> **A2.** While our approach requires re-training when introducing a new domain, we believe the associated overhead is manageable. The trainable parameters in our model-level router are significantly fewer compared to those in MoE, as mentioned in Q1.
>
> **Q3. While the approach demonstrates effectiveness compared to other quantization methods in ablation studies, the experiment results do not include a comparison with an end-to-end token-level router approach, such as MoE baselines.**
>
> **A3.** As discussed in L295-306, our method fundamentally differs from MoE. MoE employs token-level routing, which dynamically assigns individual tokens in a sequence to different experts within the same model. In contrast, our ME-Switch utilizes model-level switching, selecting the most suitable model to handle an entire user request. This distinction also significantly affects training overhead. As mentioned in Q1, training an MoE with multiple experts demands hundreds of GPU days, making a direct comparison challenging due to the limited computational resources available.
>
> **Q4. The comparison only covers up to 4 models, after which the naive implementation encounters out-of-memory (OOM) issues. It is a little subtle to conclude that latency is lower than the naive inference as the number of models is greater than 4.**
>
> **A4.** Our method scales more efficiently than the naive inference method. In the naive method, each $\mathbf{x}\mathbf{W}$ is computed independently during the forward pass, requiring a distinct $\mathbf{W}$ for each user in the batch. As the number of models grows (>=4), this approach results in substantial I/O costs due to loading of large weight matrices. In contrast, our method leverages shared pre-trained model weights $\mathbf{W}$ along with a set of small deltas $\hat{\Delta}$, significantly reducing the inference I/O burden. This is demonstrated in Figure 8 of our initial submission. We have included the discussion in the revised manuscript.

---

> ### Author Response · Authors · 2024-11-20
> **Response to Reviewer Ldho (Part 2)**
>
> **Q5. The discussion of latency mentions that the naive implementation has a slight advantage when the number of models is fewer than 4, as shown in Figure 8, and the author attributes this to the quantization overhead. However, why does the quantization overhead stop being the dominant factor as the number of models increases? What is the trend in the profiling of quantization overhead as the number of models continues to increase? A deeper analysis of this would be valuable.**
>
> **A5.** Thank you for pointing this out. Our explanation in the main text was not entirely precise. Our method is slightly slower than the naive approach when the number of models is fewer than 4 due to the additional computational cost introduced by the second term $\mathbf{x} \hat{\Delta}$ in Eq. (4). This includes the cost of I/O, dequantization, and multiplication. To illustrate this, we profile their latency percentage in Table V. We observe that the dequantization cost is very small across different model numbers. Initially, latency is dominated by I/O operations because LLM decoding is a memory-bound process when the batch size is small (Lin et al., 2024; Liu et al., 2023). However, as the number of models grows, compute-related operations, such as matrix multiplications, begin to dominate the overall latency. We have included these results and discussions in the revised manuscript.
>
> Table V. Decoding latency percentage (%) of different components for $\mathbf{x} \hat{\Delta}$. (Testing)
>
> | Model number | 1 | 2 | 4 | 8 | 16 |
> |:------------:|:--:|:--:|:--:|:--:|:--:|
> | I/O | 87.83 | 90.23 | 58.73 | 35.30 | 22.91 |
> | Dequantization | 1.74 | 0.87 | 0.15 | 0.09 | 0.05 |
> | Multiplication | 10.43 | 8.90 | 41.12 | 64.61 | 77.04 |

---

> ### Comment · Reviewer_Ldho · 2024-11-26
>
> I appreciate the authors’ detailed response. The updated explanation regarding the cost of I/O, dequantization,  and multiplication is helpful for readers to understand how latency changes as the number of models increases and the contribution of each component.
>
> The author emphasis the novelty of the paper is the efficiency improvements compared to traditional MoE methods. Overall, this work has a good vision of the drawbacks of traditional methods' efficiency and their method does show efficiency improvements over traditional token-switch approaches, particularly in reducing storage requirements, training costs, and I/O latency for large language models. However, though the paper provides performance comparisons with MoE models across up to four distinct domains, further discussion would be valuable when comparing at larger scales in a more comprehensive level. I encourage the authors to keep expanding their performance analysis relative to previous methods on more domains and more models in following works.
>
> In conclusion, I will keep my current rating.

---

> > ### Author Response · Authors · 2024-11-26
> > **Thanks for your feedback**
> >
> > Dear Reviewer Ldho,
> >
> > Thank you for your thoughtful feedback and for acknowledging the contributions and vision of our work. We appreciate your recognition of the efficiency improvements our method offers over traditional MoE approaches, as well as your thoughtful suggestions regarding larger-scale comparisons and expanded performance analysis across more domains and models.
> >
> > Your insights are helpful, and we are grateful for your time and effort in reviewing our work.
> >
> > Best regards,
> >
> > Authors of Paper #912

---

### Official Review · Reviewer_jYU9 · 2024-11-03

**Soundness:** 3
**Presentation:** 3
**Contribution:** 3
**Rating:** 6
**Confidence:** 4

**Summary:**

This paper addresses the problem of serving multiple expert models i.e multiple models finetuned for different tasks from the same base model. Serving all fully finetuned models is expensive since they take up memory and switching these in and out of memory can be slow. Existing methods use low-rank finetuning to reduce the storage required for finetuned models, however low-rank finetuning does not match full finetuning in quality hence recent methods instead quantize the delta between the finetuned model and the base model. However, these quantization methods can lead to significant quantization errors at low bitwidths. This paper proposes a saliency aware method for quantizing delta between the finetuned and base model.  In addition, this paper suggests a dynamical routing method to determine which expert model to route user requests to when the appropriate model for a user request is not known in advance.

**Strengths:**

1. This paper is generally well written and easy to follow, and the problem is well-motivated.
2. Experiments are comprehensive with convincing results and there are sufficient ablations to justify different design choices.
3. The problem of serving multiple finetuned models is a very poignant problem in this era of LLMs.

**Weaknesses:**

Major
1.  How does the saliency based delta quantization technique in this paper differ from other non-magnitude based saliency quantization techniques like https://arxiv.org/pdf/2405.14917 ?
2. Model routing is a well studied problem! Are there other existing approaches that study model routing in the same scenario where the appropriate model is not known? How is your solution different or better than them ?

Minor
1. Related work like GPT-Zip https://openreview.net/pdf?id=hO0c2tG2xL is not cited
2. On line 46, the paper claims that no single model can master all tasks simultaneously. While this is not currently the state of affairs is there a formal proof or evidence that this is impossible?

**Questions:**

1. Will the code for this be open source?
2. Are there any insights on why mixed precision quantization outperforms the full precision models in certain settings?
3. Is there an estimate of how often in practice is the appropriate model not know in advance ? Is this more common than the case where it is known?

---

> ### Author Response · Authors · 2024-11-20
> **Response to Reviewer jYU9**
>
> Thanks for your valuable comments.
>
> **Q1. How does the saliency-based delta quantization technique in this paper differ from other non-magnitude-based saliency quantization techniques like Slim-LLM [E]?**
>
> **A1.** Slim-LLM measures weight salience based on **the error induced by removing specific weights**, using the same metric as SparseGPT [F] (see Eq. (3) in [F]), which is primarily designed for pruning. In contrast, our method uses **reconstruction error (see Eq. (2)) to directly assess the impact of quantization on the model’s output**, making it more suitable for quantization tasks. To highlight the advantages of our saliency metric, we replaced it with the metric used in Slim-LLM and presented the results in Table IV. From the results, our saliency metric consistently outperforms Slim-LLM’s metric across various datasets and bitwidths. These results are included in Figure 4 and Table C of the revised manuscript.
>
> Table IV. Performance comparisons with different saliency metrics.
>
> | Domain | Dataset | BF16 | 1-bit | Slim-LLM | Ours |
> |:--------------:|:---------:|:-----:|:-----:|:--------:|:-----:|
> | Instruct (%) ↑ | MMLU | 63.43 | 63.14 | 63.26 | 63.43 |
> | Math (%) ↑ | GSM8K | 73.92 | 53.45 | 57.16 | 59.14 |
> | | Math | 20.62 | 1.50 | 2.00 | 1.74 |
> | | Avg. | 47.27 | 27.48 | 29.58 | 30.44 |
> | Code (%) ↑ | HumanEval | 51.20 | 47.00 | 48.20 | 47.60 |
> | | MBPP | 60.40 | 58.40 | 58.40 | 60.70 |
> | | Avg. | 55.80 | 52.70 | 53.30 | 54.15 |
> | Domain | Dataset | BF16 | 1-bit | Slim-LLM | Ours |
> | Instruct (%) ↑ | MMLU | 63.43 | 63.72 | 63.68 | 63.95 |
> | Math (%) ↑ | GSM8K | 73.92 | 73.31 | 72.93 | 73.62 |
> | | Math | 20.62 | 20.44 | 20.16 | 20.48 |
> | | Avg. | 47.27 | 46.88 | 46.55 | 47.05 |
> | Code (%) ↑ | HumanEval | 51.20 | 47.00 | 48.20 | 51.80 |
> | | MBPP | 60.40 | 59.10 | 59.60 | 60.70 |
> | | Avg. | 55.80 | 53.05 | 53.90 | 56.25 |
>
>
> **Q2. Model routing is a well-studied problem! Are there other existing approaches that study model routing in the same scenario where the appropriate model is not known? How is your solution different or better than them?**
>
> **A2.** Please refer to Q1 of the general response. We have moved the model-level router section to the appendix to maintain focus on our primary contribution in the main text.
>
> **Q3. Related work like GPT-Zip is not cited**
>
> **A3.** Thank you for pointing this out. We have included a citation and discussion of GPT-Zip in the related work section of the revised manuscript.
>
> **Q4. On L46, the paper claims that no single model can master all tasks simultaneously. While this is not currently the state of affairs is there formal proof or evidence that this is impossible?**
>
> **A4.** Our statement on L46 is based on practical challenges. For further details, please refer to Q2 of the general response.
>
> **Q5. Will the code for this be open source?**
>
> **A5.** Yes, we will release the code upon acceptance. We have also provided the core pseudo codes of our Triton kernel in Q13 of Reviewer 1fKA.
>
> **Q6. Are there any insights on why mixed precision quantization outperforms the full precision models in certain settings?**
>
> **A6.** The performance improvements over individual expert models are primarily attributed to our additional training through efficient distillation, as discussed in L393-395. This process improves the models’ task-specific performance by optimizing the quantization step size. Similar phenomena are also observed in many quantization literature [G][H][I].
>
> **Q7. Is there an estimate of how often in practice the appropriate model is not known in advance? Is this more common than the case where it is known?**
>
> **A7.** In public-facing applications or open-ended systems, user inputs may vary widely in content and intent, often lacking clear contextual information. This makes it particularly challenging to determine the appropriate model for queries in advance. Although it is difficult to precisely quantify how often this occurs, such scenarios are common in many practical applications.
>
> **Reference:**
>
> [E] SliM-LLM: Salience-Driven Mixed-Precision Quantization for Large Language Models. arXiv 2024.
>
> [F] SparseGPT: Massive Language Models Can be Accurately Pruned in One-Shot. ICML 2023.
>
> [G] Learned step size quantization. In ICLR, 2020.
>
> [H] Learnable Companding Quantization for Accurate Low-bit Neural Networks. CVPR 2021.
>
> [I] Nonuniform-to-Uniform Quantization: Towards Accurate Quantization via Generalized Straight-Through Estimation. In CVPR 2022.

---

> > ### Comment · Reviewer_jYU9 · 2024-11-22
> >
> > Thanks to the authors for addressing all questions. I will keep my score.

---

> > > ### Author Response · Authors · 2024-11-23
> > > **Thanks for your feedback**
> > >
> > > Dear Reviewer jYU9,
> > >
> > > Thank you for your feedback! We greatly appreciate the constructive reviews and valuable suggestions to enhance our work.
> > >
> > > Best regards,
> > >
> > > Authors of Paper #912

---

### Official Review · Reviewer_qVf8 · 2024-11-04

**Soundness:** 3
**Presentation:** 2
**Contribution:** 3
**Rating:** 5
**Confidence:** 4

**Summary:**

In this paper, the authors focus on the problem of serving multiple models (fine-tuned over
the same base) together. The challenge is that each model occupies a lot of memory and
the authors build on the idea of delta compression between the FT'ed models and the base.

Specifically, the authors proposed a mixed precision compression technique for the delta.

Moreover, the authors proposed a routing-based method to pick the right model to use
given user input.

**Strengths:**

- The proposed idea makes sense, and seems to work better than the baseline.

**Weaknesses:**

- It would be great if both contributions can be evaluated more thoroughly against previous methods
- The second contribution could be put better into context

**Questions:**

1. I am a little bit confused by how the router is related to the first contribution --
it would be great if the authors could elaborate more? these two parts look quite
orthogonal to me.

2. There are a lot of previous work (especially for the router) tackling similar problems,
it seems that the authors only compared with the most natural baseline, but not
more advanced methods. it would be great if the authors can justify it or provide more
results.

---

> ### Author Response · Authors · 2024-11-20
> **Response to Reviewer qVf8**
>
> Thanks for your constructive comments.
>
> **Q1. It would be great if both contributions can be evaluated more thoroughly against previous methods. There are a lot of previous works (especially for the router) tackling similar problems, it seems that the authors only compared with the most natural baseline, but not more advanced methods. It would be great if the authors can justify it or provide more results.**
>
> **A1.** For salient-aware delta compression, we have compared our method against various quantization techniques, as shown in Figure 4 and Table C of the initial submission. Additionally, during the rebuttal period, we conducted further evaluations using Slim-LLM’s saliency metric [E], with the results detailed in Q1 of Reviewer jYU9. These results consistently demonstrate that our method outperforms all compared approaches across different tasks and bitwidths.
>
> For further discussion on prior work related to model-level routing, please refer to Q1 of the general response.
>
> **Q2. I am a little bit confused by how the router is related to the first contribution -- it would be great if the authors could elaborate more? These two parts look quite orthogonal to me.**
>
> **A2.** Please refer to Q1 of the general response.
>
> **Q3. The second contribution could be put better into context.**
>
> **A3.** Thank you for your constructive advice. To better focus the main contribution of our method, we have moved the contents w.r.t. model-level router to the appendix. For additional details, please refer to Q1 of the general response.
>
> **Reference:**
>
> [E] SliM-LLM: Salience-Driven Mixed-Precision Quantization for Large Language Models. arXiv 2024.

---

> ### Author Response · Authors · 2024-11-24
> **Follow-up on Rebuttal**
>
> Dear Reviewer qVf8
>
> We sincerely appreciate the time and effort you have dedicated to reviewing our paper. We have carefully addressed your concerns and provided detailed responses, which we hope have resolved your queries. If you have any additional questions or further concerns, please do not hesitate to let us know.
>
> Best regards,
>
> Authors of #912

---

> ### Author Response · Authors · 2024-11-27
> **Friendly Reminder: Approaching Discussion Deadline**
>
> Dear Reviewer qVf8,
>
> We sincerely appreciate the time and effort you have dedicated to reviewing our paper. As the discussion period is nearing its end, we wanted to kindly check if our responses have sufficiently addressed your concerns. If there are any remaining issues, we would be happy to clarify further.
>
> Thank you again for your valuable feedback and time.
>
> Best regards,
>
> Authors of #912

---

### Official Review · Reviewer_1fKA · 2024-11-05

**Soundness:** 2
**Presentation:** 2
**Contribution:** 2
**Rating:** 3
**Confidence:** 4

**Summary:**

The authors introduce ME-Switch, a memory-efficient framework for serving MoE-based LLMs. Given a set of MoE models across specified domains, they present a quantization technique for deltas from the base model which they argue is better than other approaches per its retention of accuracy and effectiveness in reducing model size. The approach attempts to estimate the saliency of particular weight deltas for quantization based on a reconstruction loss-based scheme As part of the assumption that models are domain-specific, the authors train a routing model to delegate user inputs to a domain-specific model.

The authors showcase retention of performance task performance across various domains in conjunction with a reduction in GPU memory usage attributed to efficient quantization.

**Strengths:**

- The application of reconstruction-based quantization to input model weights is novel insofar as other methods (low rank approximation, uniform quantization) are not as effective.
- Results show minor to negligible losses in accuracy across many tasks, including some gains, though the degree to which these are due to noise is not immediately clear.
- The reduction in memory usage as compared to other baselines is somewhat compelling, although compared to baselines without any delta-based approaches, which the authors did not contribute.

**Weaknesses:**

- The assumption that each LLM specializes in a distinct domain is a large one, and limits the general applicability of the authors' approach. Indeed, in many MoE-based settings, model diversity is sufficient to improve performance significantly, and task specialization is not even considered. The given assumption also requires training a domain-based router, which itself requires ablations. The authors need to argue that this setting is representative and useful.
- There are missing baselines. The sensitivity and quality of the domain-based routing setup as compared to a baseline without adaptive quantization makes it difficult to disambiguate the effect of improved quantization alone.
  - Similarly, there is no baseline that foregoes SFT or one that compares SFT without quantization or task routing. These components should be independently ablated.
- Results are defined only on a very narrow set of task domains. This makes it quite unclear how the proposed approach generalizes. Domain-specific expert setups in a general setting might involve dozens of models in a large-scale system.
- In general, the paper’s writing is redundant, i.e. the core premises, contributions, and prior work are presented multiple times each, with similar levels of depth. The authors could make room to do more experiments by removing redundant components.
- The approach with which the router is trained generalizes very poorly. A router must be trained based on a collection domain-specific models for every single setup. A router trained on a specific domain-specific set of multiple choice questions is quite specific to the input tasks.
- There is no presentation of an accuracy-memory tradeoff across approaches. This, combined with weak baselines reduces the strength of the contribution.

**Questions:**

- Introduction — the authors might consider reframing motivations around MoE as well and listing memory pressure as a primary motivator for LLM task specialization.
- Line 78 the "information loss" that occurs with subsequent methods can be further specified. Is this something that affects downstream accuracy, or does this refer to something else?
- Why doesn’t reconstruction loss merely end up choosing values which are closest to values which are fp16 quantized, i.e. representable with low approximation error? More analysis of reconstruction loss would be helpful to prove efficacy.
- Why is fp16 used throughout the experiments setups rather than bf16, which is supported on A100s (used in the paper) and is far more common in inference settings?
- What is the effect of using different backbones for the same task with the proposed approach?
- What additional details can be shared about the Triton kernel? What is the performance benefit of its inclusion? How is the implementation structured?

---

> ### Author Response · Authors · 2024-11-20
> **Response to Reviewer 1fKA (Part 1)**
>
> Thanks for your constructive comments.
>
> **Q1. The assumption that each LLM specializes in a distinct domain is a large one, and limits the general applicability of the authors' approach. Indeed, in many MoE-based settings, model diversity is sufficient to improve performance significantly, and task specialization is not even considered. The authors need to argue that this setting is representative and useful.**
>
> **A1.** We would like to emphasize that our approach is designed for general-purpose LLMs rather than being tailored to MoE models. As discussed in L295-300, MoE employs **token-level routing** that assigns individual tokens in a sequence to different experts within the same model, whereas our ME-Switch uses **model-level switching** to select the most suitable model for an entire user request. For a detailed discussion regarding the assumption that each LLM specializes in a distinct domain, please refer to Q2 of the general response.
>
> **Q2. There are missing baselines. The sensitivity and quality of the domain-based routing setup as compared to a baseline without adaptive quantization makes it difficult to disambiguate the effect of improved quantization alone.**
>
> **A2.** We present the results of ME-Switch without applying our salient-aware delta compression in Tables I and II. The results demonstrate that both salient-aware delta compression alone and model-level routing alone achieve performance comparable to their respective unquantized expert models across a variety of downstream tasks. We have included the results in Section D of the revised manuscript.
>
> Table I. Performance comparison of different methods for the Mistral-7B family. "Baseline" refers to the unquantized experts, while "SADC" represents our salient-aware delta compression.
>
> | Method | STEM | Hums. | Social | Other | Avg. | GSM8K | Math | Avg. | HumanEval | MBPP | Avg. |
> |:------------------:|:-----:|:-----:|:------:|:-----:|:-----:|-------|-------|-------|-----------|-------|-------|
> | Baseline | 52.05 | 68.83 | 73.42 | 65.43 | 63.43 | 73.92 | 20.62 | 47.27 | 51.20 | 60.40 | 55.80 |
> | Router Only | 52.05 | 68.83 | 73.42 | 65.43 | 63.43 | 74.15 | 20.72 | 47.44 | 51.20 | 60.40 | 55.80 |
> | SADC Only | 53.17 | 69.09 | 73.88 | 65.40 | 63.95 | 73.62 | 20.48 | 47.05 | 51.80 | 60.70 | 56.25 |
> | SADC + Router | 51.49 | 68.37 | 73.60 | 66.08 | 63.32 | 73.39 | 20.30 | 46.85 | 51.80 | 60.70 | 56.25 |
>
> Table II. Performance comparisons of different methods for the LLaMA-2-13B family. "Baseline" refers to the unquantized experts, while "SADC" represents our salient-aware delta compression.
>
> | Method | STEM | Hums. | Social | Other | Avg. | GSM8K | Math | Avg. | C-Eval | C-MMLU | Avg. |
> |:------------------:|:-----:|:-----:|:------:|:-----:|:-----:|:-------:|:------:|:-------:|:-----------:|:-------:|:-------:|
> | Baseline | 44.26 | 59.79 | 63.20 | 56.57 | 54.60 | 69.14 | 8.48 | 38.81 | 40.28 | 39.16 | 39.72 |
> | Router Only | 44.17 | 59.73 | 63.20 | 56.57 | 54.55 |68.61 | 8.52 |  38.57 | 40.28 | 39.16 | 39.72 |
> | SADC Only | 44.57 | 60.87 | 64.00 | 58.04 | 55.45 | 70.05 | 13.20 | 41.63 | 40.13 | 39.91 | 40.02 |
> | SADC + Router | 44.51 | 60.87 | 64.00 | 58.04 | 55.43 | 69.90 | 13.14 | 41.52 | 40.13 | 39.84 | 39.99 |
>
> **Q3. There is no baseline that foregoes supervised fine-tuning (SFT) or one that compares SFT without quantization. These components should be independently ablated.**
>
> **A3.** Please refer to Q1 in the general response.

---

> ### Author Response · Authors · 2024-11-20
> **Response to Reviewer 1fKA (Part 2)**
>
> **Q4. Results are defined only on a very narrow set of task domains. This makes it quite unclear how the proposed approach generalizes. Domain-specific expert setups in a general setting might involve dozens of models in a large-scale system.**
>
> **A4.** To show how our salient-aware delta compression generalizes, we further include [BioMistral-7B](https://huggingface.co/BioMistral/BioMistral-7B) as an expert in the medical domain and [Saul-7B-Base](https://huggingface.co/Equall/Saul-7B-Base) as an expert in the legal domain. Both models are fine-tuned from [Mistral-7B-Instruct-v0.1](https://huggingface.co/mistralai/Mistral-7B-Instruct-v0.1). We evaluate their performance using subsets of MMLU corresponding to these domains. As shown in Table III, our salient-aware delta compression results in only a minor performance drop, maintaining nearly lossless performance. These results are included in Section 5.1 of the revised manuscript.
>
> Table III. Results on the medical and legal domain for the Mistral-7B family.
>
> | Method | Clinical Knowledge | Medical Genetics | Anatomy | Professional Medicine | College Biology | College Medicine | Avg. | International Law | Jurisprudence | Professional law | Avg. |
> |:------:|:------------------:|:----------------:|:-------:|:---------------------:|:---------------:|:-----------------:|:------:|:-------------------:|:---------------:|:-----------------:|:------:|
> | Baseline | 64.53 | 69.00 | 57.89 | 57.72 | 58.33 | 58.38 | 60.98 | 74.38 | 71.30 | 43.02 | 62.90 |
> | Ours | 62.26 | 68.00 | 48.89 | 57.72 | 63.89 | 61.27 | 60.34 | 75.76 | 67.59 | 43.68 | 62.34 |
>
> **Q5. In general, the paper’s writing is redundant, i.e. the core premises, contributions, and prior work are presented multiple times each, with similar levels of depth. The authors could make room to do more experiments by removing redundant components.**
>
> **A5.** Thank you for your feedback. We have simplified the writing of our manuscript, moved the router section to the appendix, and included additional experiments in the main text.
>
> **Q6. The approach with which the router is trained generalizes very poorly. A router must be trained based on a collection of domain-specific models for every single setup. A router trained on a specific domain-specific set of multiple choice questions is quite specific to the input tasks.**
>
> **A6.** Please refer to Q1 of the general response. Additionally, while it is true that the router needs to be trained for each new setup, the associated training cost is manageable due to the small number of trainable parameters, allowing it to complete within a single GPU day.
>
> **Q7. There is no presentation of an accuracy-memory trade-off across approaches.**
>
> **A7.**. We would like to clarify that Figure 4 in the initial submission presents the accuracy-memory trade-off of different methods, directly addressing this concern. The results demonstrate that our method consistently outperforms other approaches across various bitwidths and tasks.
>
> **Q8. Introduction — the authors might consider reframing motivations around MoE as well and listing memory pressure as a primary motivator for LLM task specialization.**
>
> **A8.** Thank you for the suggestion. We would like to highlight that our approach is focused on general-purpose LLMs rather than being tailored to MoE models, addressing the memory challenges associated with serving multiple specialized LLMs.
>
> **Q9. L78 the "information loss" that occurs with subsequent methods can be further specified. Is this something that affects downstream accuracy, or does this refer to something else?**
>
> **A9.** The “information loss” mentioned in L78 refers to the quantization error introduced by using per-tensor quantization (Liu et al., 2024a), as described in L76-80. This approach applies a single quantization step size ($s$ in Eq. (1)) for an entire layer, which limits its ability to capture fine-grained variations within the layer. Consequently, this quantization error can lead to a drop in final performance. To improve clarity, we have replaced the term “information loss” with “quantization error” in the revised manuscript.
>
> **Q10. Why doesn’t reconstruction loss merely end up choosing values which are closest to values which are fp16 quantized, i.e. representable with low approximation error? More analysis of reconstruction loss would be helpful to prove efficacy.**
>
> **A10.** As described in L208-215 and Eq. (2) of the manuscript, the reconstruction error is determined by both the input $\mathbf{x}\_i$ and the quantization noise $\Delta\_{ij} - \hat{\Delta}\_{ij}$ (i.e., approximation error). Even when the quantization noise is small, the reconstruction error can be significant if the input magnitude is large.

---

> ### Author Response · Authors · 2024-11-20
> **Response to Reviewer 1fKA (Part 3)**
>
> **Q11. Why is fp16 used throughout the experiment setups rather than bf16, which is supported on A100s (used in the paper) and is far more common in inference settings?**
>
> **A11.** Thank you for pointing this out. We indeed use BF16 in our experiments. This has been clarified and included in the implementation details in Section 5 of the revised manuscript.
>
> **Q12. What is the effect of using different backbones for the same task with the proposed approach?**
>
> **A12.** We would like to clarify that we have conducted experiments on the instruction domain using the Mistral-7B, LLaMA-2-13B, and LLaMA-3-8B model families. The results, presented in Tables 1 and B of the initial submission, demonstrate that our method effectively adapts to different backbones while maintaining strong performance across tasks.
>
> **Q13. What additional details can be shared about the Triton kernel? What is the performance benefit of its inclusion? How is the implementation structured?**
>
> **A13.** The Triton kernel is designed to accelerate model inference. Without it, $\tilde{\Delta}$ must be loaded from high-bandwidth memory (HBM) into SRAM, dequantized into BF16 format, written back to HBM, and then reloaded for multiplication with $\mathbf{x}$. Our Triton kernel fuses dequantization and multiplication into a single step, reducing intermediate memory operations and eliminating unnecessary data transfers, resulting in faster inference. To provide a clearer understanding, we include the core pseudo-codes of the Triton kernel as follows:

---

> ### Author Response · Authors · 2024-11-20
> **Response to Reviewer 1fKA (Part 4)**
>
> ```python
> def twobit_dequant_bmm_scale_kernel(
>     # Pointers to matrices
>     a_ptr,
>     b_ptr,
>     c_ptr,
>     scales_ptr,
>     # Matrix dimensions
>     M,
>     N,
>     K,
>     # The stride variables represent how much to increase the ptr by when moving by 1
>     # element in a particular dimension. E.g. `stride_am` is how much to increase `a_ptr`
>     # by to get the element one row down (A has M rows).
>     stride_am,
>     stride_ak,
>     stride_bk,
>     stride_bn,
>     stride_cm,
>     stride_cn,
>     stride_scales,
>     stride_batch_a,
>     stride_batch_b,
>     stride_batch_c,
>     stride_batch_scale,
>     # Meta-parameters
>     BLOCK_SIZE_M: tl.constexpr,
>     BLOCK_SIZE_N: tl.constexpr,
>     BLOCK_SIZE_K: tl.constexpr,
>     GROUP_SIZE_M: tl.constexpr,
>     ACTIVATION: tl.constexpr,
> ):
>     """Kernel for computing the matmul C = A x B.
>     A has shape (B, M, K), float
>     B has shape (B, K//n_bits, N), int, packed boolean
>     C has shape (B, M, N),
>     scales is of shape (N) float16
>     """
>     # -----------------------------------------------------------
>     # Map program ids `pid` to the block of C it should compute.
>     # This is done in a grouped ordering to promote L2 data reuse.
>     # See above `L2 Cache Optimizations` section for details.
>     pid = tl.program_id(axis=0)
>     pid_batch = tl.program_id(axis=1)
>
>     num_pid_m = tl.cdiv(M, BLOCK_SIZE_M)
>     num_pid_n = tl.cdiv(N, BLOCK_SIZE_N)
>     num_pid_k = tl.cdiv(K, BLOCK_SIZE_K)
>
>     num_pid_in_group = GROUP_SIZE_M * num_pid_n
>     group_id = pid // num_pid_in_group
>     first_pid_m = group_id * GROUP_SIZE_M
>     group_size_m = min(num_pid_m - first_pid_m, GROUP_SIZE_M)
>
>     pid_m = first_pid_m + (pid % group_size_m)
>     pid_n = (pid % num_pid_in_group) // group_size_m
>
>     offs_m = (pid_m * BLOCK_SIZE_M + tl.arange(0, BLOCK_SIZE_M)) % M
>     offs_n = (pid_n * BLOCK_SIZE_N + tl.arange(0, BLOCK_SIZE_N)) % N
>
>     offs_am = tl.max_contiguous(tl.multiple_of(offs_m, BLOCK_SIZE_M), BLOCK_SIZE_M)
>     offs_bn = tl.max_contiguous(tl.multiple_of(offs_n, BLOCK_SIZE_N), BLOCK_SIZE_N)
>     offs_k = tl.arange(0, BLOCK_SIZE_K)
>
>     a_ptrs = (
>         a_ptr
>         + (offs_am[:, None] * stride_am + offs_k[None, :] * stride_ak)
>         + pid_batch * stride_batch_a
>     )
>
>     # Adapted from GPTQ-Triton (https://github.com/fpgaminer/GPTQ-triton)
>     # b_ptrs is set up such that it repeats elements along the K axis n_bits times
>     b_ptrs = (
>         b_ptr
>         + ((offs_k[:, None] // 16) * stride_bk + offs_bn[None, :] * stride_bn)
>         + pid_batch * stride_batch_b
>     )
>     scales_ptrs = scales_ptr + offs_bn * stride_scales + pid_batch * stride_batch_scale
>
>     # (BLOCK_SIZE_K, BLOCK_SIZE_N)
>     # shifter is used to extract each bit of each element in the int matrix
>     shifter = (offs_k % 16) * 2
>     scales = tl.load(scales_ptrs)
>
>     # -----------------------------------------------------------
>     # Iterate to compute a block of the C matrix.
>     # We accumulate into a `[BLOCK_SIZE_M, BLOCK_SIZE_N]` block
>     # of bf32 values for higher accuracy.
>     # `accumulator` will be converted back to bf16 after the loop.
>     accumulator = tl.zeros((BLOCK_SIZE_M, BLOCK_SIZE_N), dtype=tl.float32)
>     for k in range(0, num_pid_k):
>         # Load the next block of A and B, generate a mask by checking the K dimension.
>         # If it is out of bounds, set it to 0.
>         a = tl.load(a_ptrs)
>         # b = tl.load(b_ptrs, mask=offs_k[:, None] < K - k * BLOCK_SIZE_K, other=0)
>         b = tl.load(b_ptrs)  # (BLOCK_SIZE_N,)
>
>         # Convert B from int to a.dtype
>         # b: (BLOCK_SIZE_K, BLOCK_SIZE_N)
>         b = (b >> shifter[:, None]) & 0x3
>         b = (b - 2).to(a.dtype)
>         b = b * scales[None, :]  # bf16
>         # b = b.to(a.dtype)
>
>         # We accumulate along the K dimension.
>         accumulator += tl.dot(a, b)
>         # Advance the ptrs to the next K block.
>         a_ptrs += BLOCK_SIZE_K * stride_ak
>         # b_ptrs += BLOCK_SIZE_K * stride_bk
>         b_ptrs += (BLOCK_SIZE_K // 16) * stride_bk
>     # You can fuse arbitrary activation functions here
>     # while the accumulator is still in bf32!
>     # if ACTIVATION == "leaky_relu":
>     #     accumulator = leaky_relu(accumulator)
>     c = accumulator.to(tl.float16)
>
>     # -----------------------------------------------------------
>     # Write back the block of the output matrix C with masks.
>     offs_cm = pid_m * BLOCK_SIZE_M + tl.arange(0, BLOCK_SIZE_M)
>     offs_cn = pid_n * BLOCK_SIZE_N + tl.arange(0, BLOCK_SIZE_N)
>     c_ptrs = (
>         c_ptr
>         + stride_cm * offs_cm[:, None]
>         + stride_cn * offs_cn[None, :]
>         + pid_batch * stride_batch_c
>     )
>     c_mask = (offs_cm[:, None] < M) & (offs_cn[None, :] < N)
>     tl.store(c_ptrs, c, mask=c_mask)
> ```
>
> We have included the above pseudo-codes in Section C of the revised manuscript.

---

> ### Author Response · Authors · 2024-11-24
> **Follow-up on Rebuttal**
>
> Dear Reviewer 1fKA
>
> We sincerely appreciate the time and effort you have dedicated to reviewing our paper. We have carefully addressed your concerns and provided detailed responses, which we hope have resolved your queries. If you have any additional questions or further concerns, please do not hesitate to let us know.
>
> Best regards,
>
> Authors of #912

---

> ### Author Response · Authors · 2024-11-27
> **Friendly Reminder: Approaching Discussion Deadline**
>
> Dear Reviewer 1fKA,
>
> We sincerely appreciate the time and effort you have dedicated to reviewing our paper. As the discussion period is nearing its end, we wanted to kindly check if our responses have sufficiently addressed your concerns. If there are any remaining issues, we would be happy to clarify further.
>
> Thank you again for your valuable feedback and time.
>
> Best regards,
>
> Authors of #912

---

### Author Response · Authors · 2024-11-20
**Response to all reviewers**

We sincerely thank all reviewers for their valuable comments.

The reviewers agree that:
### **Important problem**:
* “The problem is well-motivated … The problem of serving multiple fine-tuned models is a very poignant problem.” (Reviewer jYU9)

### **Novel and effective method**:
* “The application of reconstruction-based quantization to input model weights is novel. “ (Reviewer 1fKA)
* “The proposed idea makes sense.” (Reviewer qVf8)
* “The author proposes an innovative and effective framework for improving the storage efficiency of serving multiple LLM experts.” (Reviewer Ldho)

### **Promising performance**:
* “Results show minor to negligible losses in accuracy across many tasks, including some gains … the reduction in memory usage as compared to other baselines is somewhat compelling.” (Reviewer 1fKA)
* “Experiments are comprehensive with convincing results.” (Reviewer jYU9)
* “These experiments significantly strengthen the framework's methods ... the approach preserves accuracy competitively” (Reviewer Ldho)

# General Response

**Q1. Question regarding the model-level router.**

**A1.** We sincerely appreciate the reviewers’ feedback and would like to take this opportunity to clarify our main contribution. Our work primarily addresses the critical challenge of **high storage demands when serving multiple LLMs**. As noted in L49-71, serving three versions of LLaMA-2-70B requires over 384GB of memory, posing a substantial memory bottleneck. To tackle this issue, we propose salient-aware delta compression, which selectively quantizes the non-salient input channels of the delta weights while keeping the salient ones unchanged. This approach significantly reduces storage requirements while preserving model performance.

The model-level router is a smaller, orthogonal component included to demonstrate the feasibility of automatically selecting the optimal model for each query, given the unpredictable nature of user input. In line with Reviewer qVf8’s suggestion, we have moved the model-level router part to the appendix to maintain the focus on our primary contribution in the main text.

**Q2. Why is it assumed that no single model can master all tasks simultaneously, necessitating the use of multiple LLMs, each tailored for specific tasks?**

**A2.** The current paradigm of LLMs generally follows a pretrain-finetune framework (Achiam et al., 2023; Team et al., 2023; Touvron et al., 2023; Jiang et al., 2023). These models are first pretrained on extensive and diverse datasets to acquire broad knowledge and capabilities, and then fine-tuned on specific downstream tasks to achieve alignment or specialization. For instance, even high-capacity LLMs like the MoE model Mixtral-8x22B are fine-tuned on instruction-following data to create specialized variants such as Mixtral-8x22B-Instruct-v0.1, enhancing their ability to follow human instructions.
While LLMs are powerful, fine-tuning for a specific task to enhance performance is generally more practical and efficient than multitask fine-tuning, which often encounters conflicting objectives, mode collapse, and demands meticulous data mixing along with substantial training resources [A][B]. For example, DeepSeek-Coder-V2-Base [C], a 236B-parameter MoE code model, is fine-tuned from DeepSeek-V2 [D] to achieve significantly improved performance in the code domain (90.2% vs. 48.8% on HumanEval) but demonstrating reduced effectiveness in general question-answering tasks (47.5% vs. 53.4% on NaturalQuestions). This highlights the necessity of obtaining multiple task-specific LLMs. We have included the discussions in the introduction of the revised manuscript.

**Reference:**

[A] Llama 2: Open foundation and fine-tuned chat models. arXiv 2023.

[B] Gemini: a family of highly capable multimodal models. arXiv 2023.

[C] DeepSeek-Coder-V2: Breaking the Barrier of Closed-Source Models in Code Intelligence. arXiv 2024.

[D] DeepSeek-V2: A Strong, Economical, and Efficient Mixture-of-Experts Language Model. arXiv 2024.

# Summary of changes

We have revised our submission and summarized our updates as follows:
* We have added further discussions on our assumption that no single model can master all tasks simultaneously, highlighting the need for multiple specialized LLMs. (Reviewers 1fKA and jYU9)
* We have relocated the discussion of the model-level router to the appendix to maintain a clear focus on our primary contribution in the main text. (Reviewers 1fKA and qVf8)
* We have provided more empirical results in terms of 1) ME-Switch without salient-aware delta compression (Reviewer 1fKA); 2) more task domains (Reviewer 1fKA); 3) more saliency metric in delta compression. (Reviewer jYU9)
* We have included an additional pseudocode to illustrate the implementation of our efficient Triton kernel. (Reviewer 1fKA)
* We have updated the related work section to include a discussion of GPT-Zip. (Reviewer jYU9)

---

### Meta-Review · Area_Chair_yksc · 2024-12-20

**Metareview:**

This paper tackles the growing challenge of serving multiple fine-tuned LLMs while keeping memory usage in check. ME-Switch’s approach—using salient-aware delta compression to carefully quantize non-salient channels and preserve the important ones—feels genuinely promising. The experiments are generally well-conducted, and the authors have made a solid effort to broaden their evaluations into medical and legal domains, which is some what encouraging.

Still, certain issues remain. The scope of evaluation could be expanded, and a deeper comparison with Mixture-of-Experts (MoE) methods would better contextualize the work’s impact. While the authors’ response did offer improvements, including moving the router details to the appendix and clarfying latency considerations, some core questions about generalization and scalability remain unanswered.

Given the borderline average score, the paper’s contribution seems strong but not quite ready for acceptance. I would urge the authors to strengthen the breadth of their evaluations, incorporate MoE comparisons, and provide a clearer narrative around how these techniques can scale. With these changes, I believe this work could become a valuable contribution to the community.

**Additional Comments On Reviewer Discussion:**

The paper received split reviews (ratings from 3-6/10) with primary concerns focused on the model routing mechanism, evaluation thoroughness, and domain specialization assumptions. During rebuttal, authors clarified their main contribution was storage efficiency rather than routing (which was moved to appendix), provided additional experimental results including evaluations with new models (BioMistral-7B, Saul-7B-Base), and added detailed technical analyses. Two reviewers maintained their score while two reviewers did not change their scores.

---

### Decision · Program_Chairs · 2025-01-22

Reject